# First Hitting Diffusion Models for Generating Manifold, Graph and Categorical Data

**Mao Ye,**[*]    **Lemeng Wu,**    **Qiang Liu**
Department of Computer Science
The University of Texas at Austin

## Abstract

We propose a family of First Hitting Diffusion Models (FHDM), deep generative models that generate data with a diffusion process that terminates at a random first hitting time. This yields an extension of the standard fixed-time diffusion models that terminate at a pre-specified deterministic time. Although standard diffusion models are designed for continuous unconstrained data, FHDM is naturally designed to learn distributions on continuous as well as a range of discrete and structure domains. Moreover, FHDM enables instance-dependent terminate time and accelerates the diffusion process to sample higher quality data with fewer diffusion steps. Technically, we train FHDM by maximum likelihood estimation on diffusion trajectories augmented from observed data with conditional first hitting processes (i.e., bridge) derived based on Doob's $h$-transform, deviating from the commonly used time-reversal mechanism. We apply FHDM to generate data in various domains such as point cloud (general continuous distribution), climate and geographical events on earth (continuous distribution on the sphere), unweighted graphs (distribution of binary matrices), and segmentation maps of 2D images (high-dimensional categorical distribution). We observe considerable improvement compared with the state-of-the-art approaches in both quality and speed.

## 1  Introduction

Diffusion processes have become a powerful tool in various areas of machine learning (ML) and statistics. Traditionally, Langevin dynamics and Hamiltonian Monte Carlo have been foundations for learning and sampling from graphical models and energy-based models. Recently, denoising diffusion probabilistic models (DDPM) [18] and score matching with Langevin dynamics (SMLD) with its variants [41–43] have achieved the state-of-the-art results on data generation [13, 9, 30, 19].

Standard diffusion processes used in ML can be classified into two categories: 1) *infinite (or mixing) time* diffusion processes such as Langevin dynamics, which requires the process to run sufficiently long to converge to the *invariant distribution*, whose property is leveraged for the purpose of learning and inference; and 2) *fixed time diffusion* processes such as DDPM, SMLD, and Schrodinger bridges [11], which are designed to output the desirable results at a pre-fixed time. Although fixed-time diffusion has been show to surpass infinite time diffusion on both speed and quality, it still yield slow speed for modern applications due to the need of a pre-specified time and the incapability to adapt the time based on the difficulty of instances and problems. Moreover, standard diffusion models are naturally designed on $\mathbb{R}^d$, and can not work for discrete and structured data without special cares.

In this work, we study and explore a different *first hitting time* diffusion model that terminates at the first time as it hits a given domain, and leverages the distribution of the exit location (known as exit distribution, or harmonic measure [31]) as a tool for learning and inference. We provide the

---

[*]Corresponding author. Email: maoye21@utexas.edu

36th Conference on Neural Information Processing Systems (NeurIPS 2022).

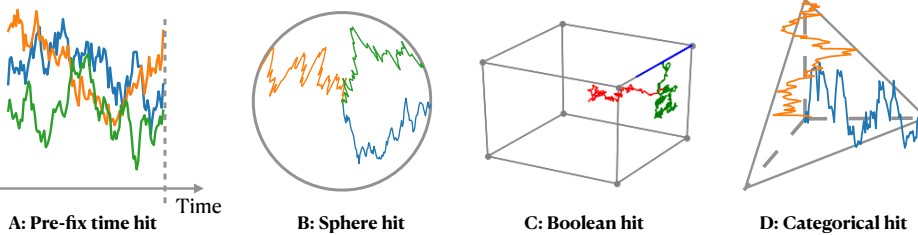

Figure 1: The four hitting schemes introduced in this paper. A: fixed-time hit, the process terminates at a fixed time; B: Sphere hit, hitting the boundary of a sphere from inside; C: Boolean hit, each coordinate terminates when it hits 0 or 1 and the whole process terminates when all of its coordinates terminate; D: Categorical hit, hitting the one-hot codes based on a conditioned process.

basic framework and tools for first hitting diffusion models. We leverage our framework to develop a general approach for learning deep generative models based on first hitting diffusion. This approach generalizes SMLD and its SDE extensions but can be attractively applied to a range of discrete and structured domains. This contrasts with the standard diffusion models, which are restricted to continuous $\mathbb{R}^d$ data. In particular, we instantiate our framework to three cases, yielding new diffusion models for learning 1) spherical, 2) binary and 3) categorical data. In addition, the proposed diffusion model gives different instances adaptive arrival times and can generate high-quality samples using fewer diffusion steps. We discuss theoretical properties and fast implementation of our methods and demonstrate their practical efficiency in a suite of practical learning problems.

## 2 Main Framework

### 2.1 First Hitting Diffusion Processes

Let $\Pi^*$ be a distribution of interest on a domain $\Omega \subset \mathbb{R}^d$. The goal is to construct a *first hitting stochastic process*, which starts from a point outside of $\Omega$ and returns a sample drawn from $\Pi^*$ when it first hits set $\Omega$. We start with introducing the new first hitting model.

Let $Z := \{Z_t \colon t \in [0, +\infty)\}$ be a continuous-time Markov process with probability law $\mathbb{Q}$ taking value in a set $V$ that contains $\Omega$ as a subset. Here $\mathbb{Q}$ is a probability measure defined on the space of all continuous trajectories $C([0, +\infty), \mathbb{R}^d)$. We use $\mathbb{Q}_t$ to denote the marginal distribution of $Z_t$ at time $t$. We assume that the process is initialized from a point $Z_0$ outside of $\Omega$. Denote by $\tau$ the *first hitting time* of $Z_t$ on $\Omega$, that is, $\tau = \inf_t \{t \geq 0 \colon Z_t \in \Omega\}$. We call that $Z_t$ is absorbing to set $\Omega$ if

i) The process enters $\Omega$ in finite time almost surely when initialized from anywhere in $V$, that is, $\mathbb{Q}(\tau < +\infty \mid Z_0 = z) = 1, \forall z \in V$.

ii) The process stops to move once it arrives at $\Omega$, that is, $\mathbb{Q}(Z_{t+s} = Z_t \mid Z_t \in \Omega) = 1, \forall s, t \geq 0$.

We define the *Poisson kernel* of $\mathbb{Q}$ as the conditional distribution of $Z_\tau$ given $Z_t = z$, denoted by $\mathbb{Q}_\Omega(\mathrm{d}x \mid Z_t = z) := \mathbb{Q}(Z_\tau = \mathrm{d}x \mid Z_t = z)$. The marginal distribution of $Z_\tau$, which we write as $\mathbb{Q}_\Omega(\mathrm{d}x) = \mathbb{Q}(Z_\tau = \mathrm{d}x)$, is called the *exit distribution, or harmonic measure*. Note that $\mathbb{Q}_\Omega(\mathrm{d}x) = \int_V \mathbb{Q}_\Omega(\mathrm{d}x \mid Z_0 = z)\mathbb{Q}_0(\mathrm{d}z)$. The crux of our framework is to leverage the exit distribution $\mathbb{Q}_\Omega$ as a tool for statistical learning and inference, which is different from traditional frameworks that exploit the properties of the distributions at a fixed time or at convergence.

**Example 2.1** (Sphere Hitting). *As shown in Figure 1-B, let $V = \{x \in \mathbb{R}^d \colon \|x\| \leq 1\}$ be the unit ball and $\Omega = S_d := \partial V$ the unit sphere. Let $Z$ be a Brownian motion starting from $z \in V$ and stopped once it hits the boundary $\Omega$. It is written as*

$$\mathbb{Q}^{S_d} \colon \qquad \mathrm{d}Z_t = \mathbb{I}(\|Z_t\| < 1)\mathrm{d}W_t, \qquad Z_0 \in V, \tag{1}$$

*where $W_t$ is a Wiener process; the indicator function $\mathbb{I}(\|Z_t\| < 1)$ sets the velocity to zero and hence stops the process once $Z_t$ hits $\Omega$. The Poisson kernel in this case is a textbook result:*

$$\mathbb{Q}_\Omega^{S_d}(\mathrm{d}x \mid Z_t = z) \propto \frac{1 - \|z\|^2}{\|x - z\|^d} \times \mu_\Omega(\mathrm{d}z), \quad \text{where } \mu_\Omega \text{ is the surface measure on } \Omega = S_d. \tag{2}$$

**Example 2.2** (Boolean Hitting). *As shown in Figure 1-C, let $V = [0,1]^d$ be the unit cube and $\Omega = B_d := \{0,1\}^d$ the Boolean cube. Let $Z$ be a Brownian motion starting from $Z_0 \in V$ and confined inside the cube $V$ in the following way:*

$$\mathbb{Q}^{B_d}: \qquad dZ_{t,i} = \mathbb{I}(Z_{t,i} \in (0,1))dW_{t,i}, \quad \forall i \in \{1,2,\cdots,d\},$$

*where $Z_{t,i}$ is the $i$-th element of $Z$. Here, each coordinate $Z_{t,i}$ stops to move once it hits one of the end points (0 or 1). It can be viewed as a particle flying in a room that sticks on a wall once it hits it.*

**Proposition 2.3.** *The Poisson kernel of $\mathbb{Q}^{B_d}$ is a simple product of Bernoulli distributions:*

$$\mathbb{Q}^{B_d}_\Omega(x \mid Z_t = z) = \mathrm{Ber}(x|z) := \prod_{i=1}^d \mathrm{Ber}(x_i|z_i), \quad \text{where} \quad \mathrm{Ber}(x_i|z_i) = x_i z_i + (1 - x_i)(1 - z_i);$$

$\mathrm{Ber}(x_i|z_i)$ *is the likelihood function of observing $x_i \in \{0,1\}$ under Bernoulli$(z_i)$ with $z_i \in [0,1]$.*

**Example 2.4** (Fixed Time Hitting). *Our first hitting framework includes the more standard models with fixed terminal time. To see this, let $\bar{Z}_t = (t, Z_t)$ be a stochastic process $Z_t$ with law $\mathbb{Q}$ augmented with time $t$ as one of its coordinates. Let $V = [0,t] \times \mathbb{R}^d$ and $\Omega = \{t\} \times \mathbb{R}^d$, where $\Omega$ is a vertical plane on the augmented space. Then the hitting time $\tau$ equals $t$ deterministically, and the exit distribution equals the marginal distribution of $Z_t$ at time $t$. See Figure 1-A, for illustration.*

## 2.2 Diffusion Process Tools: Conditioning and $h$-transform

We introduce some basic tools for diffusion processes, including how to conduct conditioning, and exponential tilting (via $h$-transform) on diffusion processes. We apply these tools to the first hitting models we have. The readers can find related background in Oksendal [31], Särkkä and Solin [37].

Assume $Z$ is a general Ito diffusion process in $V$ that is absorbed to $\Omega$, denoted as $\mathrm{Ito}_\Omega(b,\sigma)$,

$$\mathbb{Q} \sim \mathrm{Ito}_\Omega(b,\sigma): \qquad dZ_t = b_t(Z_t)dt + \sigma_t(Z_t)dW_t, \quad \forall t \in [0,+\infty), \qquad Z_0 \sim \mathbb{Q}_0, \quad (3)$$

where $b_t(x) \in \mathbb{R}^d$ is the drift term and $\sigma_t(x) \in \mathbb{R}^{d \times d}$ is a positive definite diffusion matrix. We always assume that $b$ and $\sigma$ are sufficiently regular to yield a unique weak solution of (3).

**Conditioning** A step in our work is to find the distribution of the trajectories of a process $\mathbb{Q}$ conditioned on a future event, e.g., the event of hitting a particular value $x$ at exit, that is, $\{Z_\tau = x\}$. A notable result is that the conditioned diffusion processes are also diffusion processes. Given a point $x \in \Omega$ on the exit surface, the process of $\mathbb{Q}(\cdot \mid Z_\tau = x)$ can be shown to be the law of the following diffusion process [14, 37]:

$$\mathbb{Q}(\cdot|Z_\tau = x): \quad dZ_t = \left(b_t(Z_t) + \sigma_t^2(Z_t)\nabla_{Z_t} \log q_\Omega(x \mid Z_t)\right)dt + \sigma_t(Z_t)dW_t, \quad Z_0 \sim \mu_{0|x}, \quad (4)$$

where $q_\Omega(x \mid z)$ is the density function of the Poisson kernel $\mathbb{Q}_\Omega(dx \mid Z_t = z)$ w.r.t. a reference measure $\mu_\Omega$ on $\Omega$, and $\sigma^2$ is the matrix square of $\sigma$, and the conditional initial distribution $\mu_{0|x} = \mathbb{Q}_0(\cdot \mid Z_\tau = x)$ is the posterior probability of $Z_0$ given $Z_\tau = x$.

Intuitively, the additional drift term $\nabla_{Z_t} \log p_\Omega(x \mid Z_t)$ plays the role of steering the process towards the target $x$, with an increasing magnitude as $Z_t$ approaches $\Omega$ (because $P_\Omega(\cdot \mid Z_t = z)$ converges to a delta measure centered at $x$ when $z$ approaches $\Omega$). This process is known as a diffusion *bridge*, because it is guaranteed to achieve $Z_\tau = x$ at the first hitting time with probability one.

**Proposition 2.5.** *For $\mathbb{Q}^{S_d}$, the process conditioned on $Z_\tau = x \in S_d$ at exit is*

$$\mathbb{Q}^{S_d}(\cdot \mid Z_\tau = x): \qquad dZ_t = \mathbb{I}(\|Z_t\| < 1)\left(\nabla_{Z_t} \log \frac{1 - \|Z_t\|^2}{\|x - Z_t\|^d}dt + dW_t\right). \quad (5)$$

*Here the additional drift term (colored in blue) grows to infinity if $\|Z_t\| \to 1$ but $\|Z_t - x\|$ is large, and hence enforces that $Z_\tau = x$ when we exit the unit ball.*

**Proposition 2.6.** *For $\mathbb{Q}^{B_d}$, the process conditioned on $Z_\tau = x \in \{0,1\}^d$ at exit is*

$$\mathbb{Q}^{B_d}(\cdot|Z_\tau = x): \quad dZ_{t,i} = \mathbb{I}(Z_{t,i} \in (0,1))\left(\frac{2x_i - 1}{x_i z_i + (1 - x_i)(1 - z_i)}dt + dW_{t,i}\right), \quad \forall i. \quad (6)$$

*The additional drift term (colored in blue) enforces that $Z_{\tau,i} = x_i$ at the exit time as the drift would be infinite if $z_i$ is still far from $x_i$ when $z_i$ is close to $\{0,1\}$.*

**Proposition 2.7.** *For the fixed time diffusion in Example 2.4, let $\mathbb{Q}^T$ be the standard Brownian motion $dZ_t = dW_t$ stopped at a fixed time $t = T$, then $\mathbb{Q}$ conditioned on $\mathbb{Q}^T(Z|Z_T = x)$ is*

$$\mathbb{Q}^T(\cdot|Z_\tau = x): \qquad dZ_t = \mathbb{I}(t \leq T)\left(\frac{Z_t - x}{T - t}dt + dW_t\right). \qquad (7)$$

*The additional drift (colored in blue) forces $Z_T = x$ as it grows to infinity if $Z_t \neq x$ while $t \to T$.*

$h$**-Transform**    Assume we want to modify the Markov process $Z$ such that its exit distribution $\mathbb{Q}_\Omega$ matches the desirable target distribution $\Pi^*$. Doob's $h$-transform [14] provides a simple general procedure to do so. Note that by disintegration theorem, we have $\mathbb{Q}(dZ) = \int \mathbb{Q}_\Omega(dx)\mathbb{Q}(dZ \mid Z_\tau = x)$, which factorizes $\mathbb{Q}$ into the product of the exit distribution and the conditional process given a fixed exit location $Z_\tau = x$. To modify the exit distribution of $\mathbb{Q}$ to $\Pi^*$, we can simply replace $\mathbb{Q}_\Omega$ with $\Pi^*$ in the disintegration theorem, yielding

$$\mathbb{Q}^{\Pi^*}(dZ) := \int \Pi^*(dx)\mathbb{Q}(dZ \mid Z_\tau = x) = \pi^*(Z_\tau)\mathbb{Q}(dZ), \quad \text{with} \quad \pi^*(Z_\tau) := \frac{d\Pi^*}{d\mathbb{Q}_\Omega}(Z_\tau), \quad (8)$$

where $\pi^* = \frac{d\Pi^*}{d\mathbb{Q}_\Omega}$ is the Radon–Nikodym derivative (or density ratio) between $\Pi^*$ and $\mathbb{Q}_\Omega$, and $\mathbb{Q}^{\Pi^*}$ is called an $h$-transform of $\mathbb{Q}$. Intuitively, $\mathbb{Q}^{\Pi^*}$ is the distribution of trajectories $Z \sim \mathbb{Q}(\cdot|Z_\tau = x)$ when the exit location $x$ is randomly drawn from $x \sim \Pi^*$. We can also view $\pi^*(Z_\tau)$ as an importance score of each trajectory $Z$ based on its terminal state $Z_\tau$, and $\mathbb{Q}^{\Pi^*}$ is obtained by reweighing (or tilting) the probability of each trajectory based on its score.

If $\mathbb{Q}$ is a diffusion process, then $\mathbb{Q}^{\Pi^*}$ is also a diffusion process. In addition, $\mathbb{Q}^{\Pi^*}$ is the law of the following diffusion process:

$$\mathbb{Q}^{\Pi^*}: \qquad dZ_t = \left(b_t(Z_t) + \sigma_t^2(Z_t)\nabla_z \log h_t^{\Pi^*}(Z_t)\right)dt + \sigma_t(Z_t)dW_t, \quad Z_0 \sim \mathbb{Q}_0^{\Pi^*} \qquad (9)$$

where the initial distribution $\mathbb{Q}_0^{\Pi^*}$ and $h^{\Pi^*}$ in the drift term are defined as

$$\mathbb{Q}_0^{\Pi^*}(dz) = \int_\Omega \pi^*(x)\mathbb{Q}(Z_\tau = dx, Z_0 = dz) \qquad (10)$$

$$h_t^{\Pi^*}(z) = \mathbb{E}_\mathbb{Q}[\pi^*(Z_\tau) \mid Z_t = z] = \int_\Omega \pi^*(x)\mathbb{Q}(Z_\tau = dx \mid Z_t = z). \qquad (11)$$

It is clear that $h$ coincides with $\pi^*$ on the boundary, that is, $h_{\pi^*}(x, t) = \pi^*(x)$ for all $x \in \Omega, t \geq 0$. The name of $h$-transform comes from the fact that $h^{\Pi^*}$ is a (space-time) harmonic function w.r.t. $\mathbb{Q}$ in the light of a mean value property: $h_t^{\Pi^*}(z) = \mathbb{E}_\mathbb{Q}[h_{t+s}^{\Pi^*}(Z_{t+s}) \mid Z_t = z], \forall s, t > 0$. $\mathbb{Q}^{\Pi^*}$ yields a simple variational representation in terms of Kullback–Leibler (KL) divergence.

**Proposition 2.8** (Variational Principle). *The $\mathbb{Q}^{\Pi^*}$ in (8) yields*

$$\mathbb{Q}^{\Pi^*} = \underset{\mathbb{P}\in\mathcal{P}(V,\Omega)}{\arg\min}\left\{\mathcal{KL}(\mathbb{P} \mid\mid \mathbb{Q}) := \mathbb{E}_\mathbb{P}\left[\log\frac{d\mathbb{P}}{d\mathbb{Q}}(Z)\right], \quad s.t. \quad \mathbb{P}_\Omega = \Pi^*\right\} \qquad (12)$$

$$= \underset{\mathbb{P}\in\mathcal{P}(V,\Omega)}{\arg\min}\left\{\mathcal{KL}(\mathbb{P} \mid\mid \mathbb{Q}^{\Pi^*}) \equiv \mathcal{KL}(\mathbb{P} \mid\mid \mathbb{Q}) - \mathbb{E}_\mathbb{P}[\log \pi^*(Z_\tau)]\right\}, \qquad (13)$$

*where $\mathcal{P}(V, \Omega)$ denotes the set of path measures on $V$ that is absorbed to $\Omega$.*

Eq. (12) shows that $\mathbb{Q}^{\Pi^*}$ is the distribution with $\Pi^*$ as the exit distribution that has the minimum KL divergence with $\mathbb{Q}$. It can be viewed as a Schrodinger half bridge problem [e.g., 32], which enforces the constraint of $\mathbb{P}_T = \Pi^*$ at a fixed time $T$, rather than the first hitting time $\tau$. Eq. (13) shows that the constraint can be turned into a penalty.

**First Hitting Diffusion for Sampling**    The $h$-transform above readily provides a first hitting diffusion approach to approximate sampling from $\Pi^*$, assuming we can approximate the drift term $h^{\Pi^*}$. The Schrodinger-Follmer sampler [20] can be viewed as a special case of this approach with a fixed exit time. We leave further exploration to future works. See more discussion in Appendix A.3.

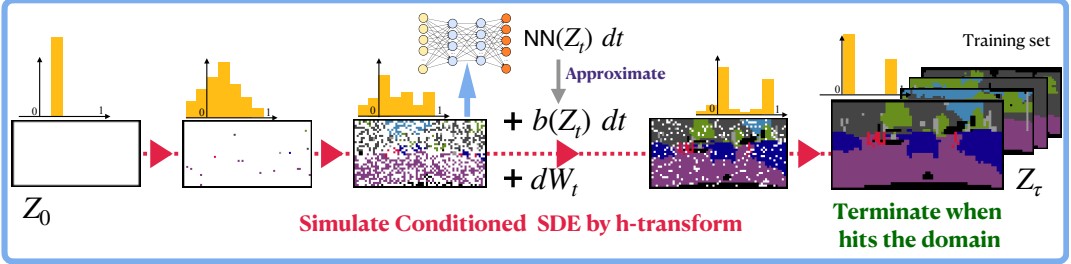

Figure 2: The training pipeline of FHDM. Start from initial distribution, we use h-transform to simulate a conditioned SDE such that the process terminates at the desired destination data from training set at its hitting time. The network is trained to approximate the drift term ($b(Z_t)$), resulting a score-matching loss that is equivalent to the KL divergence.

---

**Algorithm 1** Learning Generative Models by First Hitting Diffusion

---

**Inputs & Goal**: A data $\hat{\Pi} := \{x^{(i)}\}$ drawn from $\Pi^*$ on $\Omega$. A baseline process $\mathbb{Q}$ and a model $\mathbb{P}^\theta$ that are absorbing to $\Omega$. Want to find $\theta$ such that $\mathbb{P}_\Omega^\theta \approx \Pi^*$.
**Training**: Approximate $\hat{\theta} = \arg\min_\theta \mathcal{L}(\theta)$ by stochastic gradient descent with batches of data (approximately) drawn $Z \sim \mathbb{Q}(\cdot|Z_\tau = x)$ and $x \sim \hat{\Pi}$.
**Inference**: Simulate $\mathbb{P}^{\hat{\theta}}$.

---

## 2.3 Learning First Hitting Diffusion Models

We illustrate the learning pipeline of our First Hitting Diffusion Models (FHDM) in Figure 2. Assume $\Pi^*$ is unknown and we observe it through an i.i.d. sample $\{x^{(i)}\}_{i=1}^n$ drawn from $\Pi^*$. We want to fit the data with a parametric diffusion process $\text{Ito}_\Omega(s_\theta, \sigma)$ in $V$ that is absorbing to $\Omega$,

$$\mathbb{P}^\theta: \qquad\qquad \mathrm{d}Z_t = s_t^\theta(Z_t)\mathrm{d}t + \sigma_t(Z_t)\mathrm{d}W_t, \qquad\qquad Z_0 \sim \mathbb{P}_0^\theta, \qquad (14)$$

such that the exit distribution $\mathbb{P}_\Omega^\theta$ matches the unknown $\Pi^*$. Here $s_t^\theta(z)$ is a deep neural network with input $(z, t)$ and parameters $\theta$. We should design $s^\theta$ and $\sigma$ properly to ensure the absorbing property. The standard approach to estimate $\Pi^*$ is maximum likelihood estimation, which can be viewed as approximately solving $\min_\theta \mathcal{KL}(\Pi^* \parallel \mathbb{P}_\Omega^\theta)$. However, calculating the likelihood of the exit distribution $\mathbb{P}_\Omega^\theta$ of a general diffusion process is computationally intractable. To address this problem, we fix $\mathbb{Q}$ as a "prior" process, and augment the data distribution $\Pi^*$ to the $h$-transform $\mathbb{Q}^{\Pi^*}$, whose exit distribution $\mathbb{Q}_\Omega^{\Pi^*}$ matches $\Pi^*$ by definition. Note that we can draw i.i.d. sample from $\mathbb{Q}^{\Pi^*}$ in a "backward" way: first drawing an exit location $x \sim \Pi^*$ from the data, and then draw the trajectory $Z$ from $\mathbb{Q}(\cdot|Z_\tau = x)$ with the fixed exit point. To train a generative model, we train $\mathbb{P}^\theta$ to fit it with the data drawn from $\mathbb{Q}^{\Pi^*}$ by maximum likelihood estimation:

$$\min_\theta \left\{ \mathcal{L}(\theta) := \mathcal{KL}(\mathbb{Q}^{\Pi^*} \parallel \mathbb{P}^\theta) \equiv -\mathbb{E}_{Z\sim\mathbb{Q}^{\Pi^*}}\left[\log p^\theta(Z)\right] + const, \right\},$$

where $p^\theta = \frac{\mathrm{d}\mathbb{P}^\theta}{\mathrm{d}\mathbb{Q}^{\Pi^*}}$ is Radon–Nikodym density function of $\mathbb{P}^\theta$ relative to $\mathbb{Q}^{\Pi^*}$. By the chain rule of KL divergence in (20) in Appendix A.9, we have $\mathcal{KL}(\Pi^* \parallel \mathbb{P}_\Omega^\theta) \leq \mathcal{KL}(\mathbb{Q}^{\Pi^*} \parallel \mathbb{P}^\theta)$. Therefore, if minimizing the KL divergence allows us to achieve $\mathbb{P}^\theta \approx \mathbb{Q}^{\Pi^*}$, we should also have $\mathbb{P}_\Omega^\theta \approx \mathbb{Q}_\Omega^{\Pi^*} = \Pi^*$.

Using Girsanov theorem [24], we can calculate the density function $p^\theta$ and hence the loss function.

**Proposition 2.9.** *Assume $\mathbb{Q}$ in (3), and $\mathbb{P}^\theta$ in (14) are absorbing to $\Omega$. We have*

$$\mathcal{L}(\theta) = \frac{1}{2}\mathbb{E}_{\mathbb{Q}^{\Pi^*}}\left[\int_0^\tau \left\|\sigma_t(Z_t)^{-1}(s_t^\theta(Z_t) - b_t(Z_t \mid Z_\tau))\right\|^2 \mathrm{d}t - \log p_0^\theta(Z_0)\right] + const, \qquad (15)$$

*where $b_t(z|x) := b_t(z) + \sigma_t^2(z)\nabla_z \log p_\Omega(x|z)$ is the drift of the conditioned process $\mathbb{Q}(\cdot|Z_\tau = x)$ in (4), and $p_0^\theta$ is the probability density function of the initial distribution $\mathbb{P}_0^\theta$. In addition, $\theta^*$ achieves the global minimum of $\mathcal{L}(\theta)$ if*

$$s_t^{\theta^*}(z) = \mathbb{E}_{Z\sim\mathbb{Q}^{\Pi^*}}[b_t(z|Z_\tau) \mid Z_t = z], \qquad\qquad \mathbb{P}_0^{\theta^*} = \mathbb{Q}_0^{\Pi^*} = \mathbb{E}_{x\sim\Pi^*}[\mathbb{Q}_0^x(\cdot)].$$

Therefore, the optimal drift term $s_t^{\theta^*}$ should match the conditional expectation of $b_t(z|x)$ with $x \sim \mathbb{Q}_\Omega(\cdot|Z_t = z)$, which coincides with the drift of $\mathbb{Q}^{\Pi^*}$ in (9). The initial distribution of $\mathbb{P}^\theta$ should obviously match the initial distribution of $\mathbb{Q}^{\Pi^*}$. In practice, we recommend eliminating the need of estimating $\mathbb{P}^{\theta_0}$ by starting $\mathbb{Q}$ from a deterministic point $Z_0 = z_0$, in which case $\mathbb{P}^\theta$ should initialize from the same deterministic point. See Algorithm 1.

**Learning Spherical Hitting Models**    Take $\mathbb{Q} = \mathbb{Q}^{S_d}$ in Example 2.1, we get a method for learning generative models for data on the unit sphere. We set the model to be $\mathrm{d}Z_t = \mathbb{I}(\|Z_t\| < 1)(f_t^\theta(Z_t)\mathrm{d}t + \mathrm{d}W_t)$ to ensure that it is absorbing to $S_d$. The loss function is

$$\mathcal{L}(\theta) = \frac{1}{2}\mathbb{E}_{\substack{x \sim \Pi^* \\ Z \sim \mathbb{Q}^x}}\left[\int_0^\tau \left\|f_t^\theta(Z_t) - \nabla_{Z_t}\log\frac{1 - \|Z_t\|^2}{\|x - Z_t\|^d}\right\|^2 \mathrm{d}t - \log p_0^\theta(Z_0)\right] + const.$$

**Learning Boolean Hitting Models**    Taking $\mathbb{Q} = \mathbb{Q}^{B_d}$ as in Example 2.2 provides an approach to learning diffusion generative models for binary variables. We set the model $\mathbb{P}^\theta$ to be $\mathrm{d}Z_t = \mathbb{I}(Z_t \in (0,1)) \circ (f_t^\theta(\theta)\mathrm{d}t + \mathrm{d}W_t)$ to ensure that $\mathbb{P}^\theta$ is absorbing to $B_d$ like $\mathbb{Q}^{B_d}$, where $\circ$ denotes element-wise multiplication. The loss function is

$$\mathcal{L}(\theta) = \frac{1}{2}\mathbb{E}_{\substack{x \sim \Pi^* \\ Z \sim \mathbb{Q}^x}}\left[\int_0^\tau \left\|\mathbb{I}(Z_t \in (0,1)) \circ \left(f_t^\theta(Z_t) - \nabla_{Z_t}\log\mathrm{Ber}(Z_t|x)\right)\right\|^2 \mathrm{d}t - \log p_0^\theta(Z_0)\right] + const.$$

**Learning Fixed Time Diffusion Models**    Following the fixed time setting in Example 2.4, we can recover the standard fixed time diffusion models for continuous data, such as SMLD and DDPM. In particular, a natural choice is to set $\mathbb{Q}$ to be an O-U process $\mathrm{d}Z_t = \alpha_t Z_t \mathrm{d}t + \sigma_t \mathrm{d}W_t$ initialized from $Z_0 \sim \mathcal{N}(0, v_0)$ where $\sigma_t \geq 0, v_0 > 0$. We show in Appendix A.4 that SMLD ($\alpha_t = 0$) and DDPM ($\alpha_t > 0$) is recovered as the limit case when $v_0 \to +\infty$.

### 2.3.1   Learning Categorical Generative Models

In addition to the boolean hitting model, we provide here a first hitting framework for learning categorical data. In this case, the data domain $\Omega$ is $C_{d,m} = \{e_1, \ldots, e_d\}^m$, where $e_i = [0, \ldots, 1, \ldots, 0]$ is the $i$-th one-hot (or basis) vector in $\mathbb{R}^d$, so the data is a $m$-dimensional and $d$-categorical.

It is less straightforward to construct a first hitting diffusion process that is absorbing to $C_{d,m}$. We leverage the conditioning technique to achieve this. We explain the idea with $m = 1$, of which the general case is a direct product. The key observation is that the one-hot vectors $C_{d,1}$ is a subset of the boolean cube $B_d = \{0,1\}^d$. Hence, by definition, the conditioned process $\mathbb{Q}^{C_{d,1}} := \mathbb{Q}^{B_d}(\cdot|Z_\tau \in C_{d,1})$ exits at $C_{d,1}$ from the inside of $B_d$. Using the method of $h$-transforms [14, 37], $\mathbb{Q}^\Omega := \mathbb{Q}^{B_d}(\cdot|Z_\tau \in \Omega)$ for any $\Omega \subset B_d$ is the law of

$$\mathrm{d}Z_t = \mathbb{I}(Z_t \in (0,1)) \circ (\nabla_z \log\mathrm{Ber}(\Omega \mid Z_t)\mathrm{d}t + \mathrm{d}W_t), \qquad \mathrm{Ber}(\Omega \mid z) := \textstyle\sum_{e \in \Omega}\mathrm{Ber}(e \mid z).$$

Another challenge is to construct a parametric family of $\mathbb{P}^\theta$ that is absorbing to $C_{d,m}$, regardless of the value of $\theta$. The result below shows that this can be done by simply adding on top of $\mathbb{Q}^\Omega$ any bounded neural network drift term.

**Proposition 2.10.** *Let $V = [0,1]^d$ and $\Omega$ is any subset of $B_d = \{0,1\}^d$. Assume $f_t^\theta(z)$ is any bounded measurable function. Then the following process is guaranteed to hit $\Omega$ when it exits $V$:*

$$\mathrm{d}Z_t = \mathbb{I}(Z_t \in (0,1)) \circ \left(f_t^\theta(Z_t) + \nabla_z \log\mathrm{Ber}(\Omega \mid Z_t)\mathrm{d}t + \mathrm{d}W_t\right), \quad Z_0 \in (0,1)^d.$$

See Appendix A.1 for the summary of the algorithm for learning categorical data.

### 2.3.2   Fast Sampling of Bridges

One main step in calculating the loss $\mathcal{L}(\theta)$ is to draw trajectory $Z$ from the bridge $\mathbb{Q}^x = \mathbb{Q}(\cdot \mid Z_\tau = x)$. This can be achieved by simulating the bridge processes using Euler–Maruyama method. This is not computationally costly because it is the simulation of elementary SDEs and does not involve deep neural networks. However, it does cause a slow down in the training algorithm if the data $x$ is very high dimensional and the data size is very large.

To speed up the training, we propose a fast algorithm for simulating bridges by exploiting the symmetry when we initialize from a point $z_0$ (e.g., the center of sphere) around which $\Omega$ and $\mathbb{Q}$ are rotational symmetric. The idea is simple: we simulate the *unconditioned* process $\mathbb{Q}$ to get a trajectory $Z$ that exits at any point. Then, to obtain the conditional process $\mathbb{Q}(\cdot|Z_\tau = x)$, we simply rotate the trajectory $Z$ such that the exit point $Z_\tau$ is transformed from the original one to $x$. An advantage is that we can pre-simulate a large number of trajectories before training and only need to apply the rotation operator to get specific conditioned processes during training. Figure 3 gives an illustration using the example of sphere hit. The idea can be applied similarly for other types of hitting.

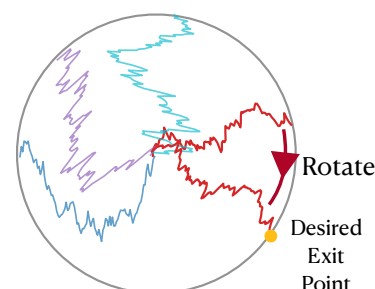

Figure 3: To sample a conditioned process, we can pick up a trajectory of unconditioned process and rotate it so that it exits at a given point.

**Proposition 2.11.** *Assume $Z$ with law $\mathbb{Q}$ initialized from $z_0 \in \mathbb{R}^d$ is absorbing to $\Omega$. For $x, x' \in \Omega$, let $\mathtt{rot}_{x' \to x}$ be the rotation operator around $z_0$ that transforms $x'$ to $x$ (hence $\mathtt{rot}_{x' \to x}(x') = x$). Assume that $\Omega$ and $\mathbb{Q}$ are rotation invariant around $z_0$ in that $\mathtt{rot}_{x',x}(\Omega) = \Omega$ and $\mathtt{rot}_{x',x}(Z) \sim \mathbb{Q}$ when $Z \sim \mathbb{Q}$ for any $x', x \in \Omega$. Then if $Z \sim \mathbb{Q}$, we get $Z' = \mathtt{rot}_{Z_\tau \to x}(Z)$, a sample drawn from $\mathbb{Q}(\cdot|Z_\tau = x)$.*

Such a fast sampling approach is applicable for all the categorical, sphere and binary distributions.

## 2.4 Discretization Error

In practice, the Euler-Maruyama method is applied to discretize the process. Analyzing the discretization error of a random hitting process is more difficult than that of a fixed-time process. In a fixed-time process, both discretized and continuous processes terminate at the same time, making the coupling tricks applicable for analyzing the discretization error based on the $\ell_2$ Wasserstein distance. Standard analysis under Lipschitz continuity assumption of the drifts gives $O(\Delta)$ error rate where $\Delta$ is the discretization step size [37]. In comparison, the key challenge of analyzing the FHDM is that the discretized and continuous processes may not terminate at the same time, and thus we need to bound the probability of the difference of the hitting time distribution in the analysis. Besides, in practice, we might also apply some time truncation tricks in order to have a bounded waiting time for generating. In Appendix A.8, we provide a full analysis and show that FHDM also yields $O(\Delta)$ discretization error asymptotically.

## 3 Related Work

**Diffusion Generative Model on Different Domains**  Diffusion generative model has been demonstrated to be powerful in generation of general continuous data such as image [40, 41, 18, 42, 43, 13], point cloud shape [7, 26, 49] and audio [9, 22]. Recently, diffusion generative model has also been extended to learn to generate data on special domains such graph [30], segmentation map [19], text [19] and manifold data [12]. Such a generalization of diffusion model is usually case-by-case and is based on applying constraints to ensure the data remains in the desired domain during the diffusion process [19, 12] or use heuristic approximation to round the data into the discrete space [30]. Our FHDM gives a unified framework for generating data on special domain via a completely new mechanism of first hitting.

**Theoretical Framework on Diffusion Process**  Most existing diffusion models are based on the framework of time-reversing [43] in which the generation (i.e. denoising) process is learned based on its time-reversed stochastic differential equation trajectory that can be simulated easily, ignoring the mismatch of the initial distribution. In comparison, our framework is conceptually simpler and is only based on a forward process, in which the learning is based on conditioned stochastic differential equations (i.e., bridge) that can be simulated via $h$-transform. A similar framework is independently explored in [34] but our method is more general and exploits the idea of first hitting. Schrodinger bridges is another well studied framework of diffusion model [46, 11, 34, 10]. However, using Schrodinger bridges usually require expensive forward-backward algorithms. It is also unknown whether or how Schrodinger bridges can be applied for generating data in special domains.

| Model | Airplane | | | | Chair | | | |
|---|---|---|---|---|---|---|---|---|
| | MMD↓ | COV↑ | 1-NNA↓ | JSD↓ | MMD↓ | COV↑ | 1-NNA↓ | JSD↓ |
| PC-GAN [1] | 3.819 | 42.17 | 77.59 | 6.188 | 13.436 | 46.23 | 69.67 | 6.649 |
| GCN-GAN [45] | 4.713 | 39.04 | 89.13 | 6.669 | 15.354 | 39.84 | 77.86 | 21.71 |
| Tree-GAN [38] | 4.323 | 39.37 | 83.86 | 15.646 | 14.936 | 38.02 | 74.92 | 13.28 |
| PointFLow [47] | 3.688 | 44.98 | 66.39 | 1.536 | 13.631 | 41.86 | 66.13 | 12.47 |
| ShapeGF [7] | 3.306 | **50.41** | **61.94** | 1.059 | 13.175 | 48.53 | **56.17** | 5.996 |
| DPM[26] | **3.276** | 48.71 | 64.83 | 1.067 | 12.276 | 48.94 | 60.11 | 7.797 |
| Ours | 3.350 | **50.41** | 67.21 | **0.986** | **6.644** | 49.50 | 56.87 | **5.913** |

Table 1: Result of point cloud generation experiment. We adopt the base line from Luo and Hu [26]. Bolded value indicates the best performance method.

## 4 Experiments

We applied FHDM to distributions on various domains such as point cloud (general continuous distribution), distribution of climate and geography events on earth (continuous distribution on the sphere), unweighted graphs (distribution of binary matrices), and segmentation map of 2D image (high dimension categorical distribution). We demonstrate that

• 1. As a generalization of the fixed-time processes such as DDPM, the fixed-time scheme of FHDM is a generative model of higher quality for general continuous distribution (section 4.1).

• 2. As a versatile model, FHDM is able to learn the distribution in many different domains and it outperforms existing specifically designed generative models (see section 4.1).

• 3. The hitting time of FHDM is well-bounded and in several tasks, FHDM even requires much fewer diffusion steps than existing methods while generating higher quality samples (see section 4.2).

Besides, we also conduct experiments to understand the intuition of the first time hitting mechanism (section 4.2) and demonstrate the acceleration of the fast sampling approach introduced in Section 2.3.2 (see in Appendix A.6). We include the visualization of the generated samples in Appendix A.7. Please find the code at https://github.com/lushleaf/first_hitting_diffusion.

### 4.1 Generation Experiment

**Point Cloud Generation** Following Luo and Hu [26], we employ the ShapeNet dataset [8] to evaluate the generated point cloud. We compare our approach against several the state-of-the-art generative models including PC-GAN [1], GCN-GAN [45], Tree-GAN [38], PointFlow [47], ShapeGF [7] and DPM [26]. See Appendix A.5 for training details. Following Cai et al. [7], Luo and Hu [26], we use minimum matching distance (MMD) and the coverage score (COV) paired with Chamfer distance as well as 1-NN classifier accuracy and the Jenson-Shannon divergence (JSD) to evaluate the quality of the generated point cloud. We refer readers to Appendix A.5 for more details on the metrics. Same to Cai et al. [7], Luo and Hu [26], we evaluate the quality on two categories, Airplane and Chair and the generated and reference point clouds are normalized into a bounding box of $[-1, 1]^3$ at evaluation. Table 1 summarizes the results showing that FHDM achieves the best performance on most criterion.

**Generating Distribution on Sphere** We apply FHDM to generate distribution of occurrences of earth and climate science events on the surface of earth (which is approximated as a perfect sphere). Following De Bortoli et al. [12], we consider 4 datasets: volcanic eruption [29], earthquakes [28], floods [5] and wild fires [15]. We compared FHDM against the current the state-of-the-art baselines including Riemannian Continuous Normalizing Flows [27], Moser Flows [36], mixture of Kent distributions [33] and standard Score-Based Generative model on 2D plane followed by the inverse stereographic projection (Stereographic Score-Based) [16] and Riemannian Generative Model [12]. Same to De Bortoli et al. [12], we evaluate the method via the negative log-likelihood on the test set. We run our method for 5 independent trials and report the averaged metric with its standard deviation. We directly adopt the baseline result from De Bortoli et al. [12]. Table 2 summarizes the result. See Appendix A.5 for additional details.

|  | Volcano | Earthquake | Flood | Fire |
|---|---|---|---|---|
| Mixture of Kent [33] | $-0.80 \pm 0.47$ | $0.33 \pm 0.05$ | $0.73 \pm 0.07$ | $-1.18 \pm 0.06$ |
| Riemannian CNF [27] | $-0.97 \pm 0.15$ | $0.19 \pm 0.0.4$ | $0.90 \pm 0.03$ | $-0.66 \pm 0.05$ |
| Moser Flow [36] | $-2.02 \pm 0.42$ | $-0.09 \pm 0.02$ | $0.62 \pm 0.04$ | $-1.03 \pm 0.03$ |
| Stereographic Score-based [16] | $-4.18 \pm 0.30$ | $-0.04 \pm 0.11$ | $1.31 \pm 0.16$ | $0.28 \pm 0.20$ |
| Riemannian Score-based [12] | $\mathbf{-5.56 \pm 0.26}$ | $-0.21 \pm 0.03$ | $0.52 \pm 0.02$ | $\mathbf{-1.24 \pm 0.07}$ |
| Ours | $-1.25 \pm 0.18$ | $\mathbf{-0.27 \pm 0.02}$ | $\mathbf{0.29 \pm 0.03}$ | $-1.24 \pm 0.08$ |

Table 2: Result on generating distribution of occurrences of earth and climate science events on the surface of earth. Bolded value indicates the best method.

| Method | Community-small | | | | Ego-small | | | | Avg. |
|---|---|---|---|---|---|---|---|---|---|
|  | Deg. | Clus. | Orbit. | Avg. | Deg. | Clus. | Orbit. | Avg. | |
| GraphVAE [39] | 0.350 | 0.980 | 0.540 | 0.623 | 0.130 | 0.170 | 0.050 | 0.117 | 0.370 |
| DeepGMG [23] | 0.220 | 0.950 | 0.400 | 0.523 | 0.040 | 0.100 | 0.020 | 0.053 | 0.288 |
| GraphRNN [48] | 0.080 | 0.120 | 0.040 | 0.080 | 0.090 | 0.220 | 0.003 | 0.104 | 0.092 |
| GNF [25] | 0.200 | 0.200 | 0.110 | 0.170 | 0.030 | 0.100 | **0.001** | 0.044 | 0.107 |
| EDP-GNN[30] | 0.053 | 0.144 | 0.026 | 0.074 | 0.052 | 0.093 | 0.007 | 0.050 | 0.062 |
| Ours | **0.009** | **0.105** | **0.009** | **0.041** | **0.019** | **0.040** | 0.005 | **0.021** | **0.031** |

Table 3: Result on graph generation experiment. We report the averaged performance of our approach based on 5 independent runs, giving 0.0013 standard deviation of the averaged metric. The results of the other baselines are directly adopted from Niu et al. [30]. Bolded value indicates the best method.

**Graph Generation**   We apply FHDM to generate (unweighted) graph that can be represented using binary adjacency matrix. Following the experiment setup in You et al. [48], Liu et al. [25], Niu et al. [30], we compare methods on two widely used benchmark datasets, Community-small and Ego-small. We apply the EDP-GNN [30] that preserves the node permutation invariance to approximate the drift. We compare FHDM against GraphRNN [48], GNF [25], GraphVAE [39] and DeepGMG [23]. The maximum mean discrepancy (MMD) over three graph statistics (1. degree distribution; 2. cluster coefficient distribution; 3. the number of orbits with 4 nodes) proposed by You et al. [48] is used to evaluate the quality of the generative graphs. For our approach, we run 5 independent trails and report the averaged performance. See Appendix A.5 for additional training details. Table 3 summarizes the result, suggesting considerable improvement over the baselines.

**Segmentation Map Generation**   FHDM can also be applied to generate high dimensional categorical distribution such as the segmentation map of a 2D image. Following Hoogeboom et al. [19], we aim to learn a model to generate the segmentation map of cityscapes dataset, in which the value of each pixel represents the category of the object that pixel belongs to. Following the setup in Hoogeboom et al. [19], there are in total 8 categories and the value at each pixel is coded using one-hot vector. We compare our

| Method | ELBO | IWBO |
|---|---|---|
| Round / Unif [44] | 1.010 | 0.930 |
| Round / Var [17] | 0.334 | 0.315 |
| Argmax / Softplus thres. [19] | 0.303 | 0.290 |
| Argmax / Gumbel dist. [19] | 0.365 | 0.341 |
| Argmax / Gumbel thres. [19] | 0.307 | 0.287 |
| Multinomial Diffusion [19] | 0.305 | - |
| Ours | **0.066** | **0.065** |

Table 4: Result for segmentation map generation. We run our method for 5 independent runs and report the averaged performance. FHDM gives 0.003/0.006 standard deviation of ELBO/IWBO.

approach with uniform dequantization [44], variational dequantization [17], three variants of argmax flow [19] and multinomial diffusion [19]. Following Hoogeboom et al. [19], we evaluate the quality of generative model by evidence lower bound (ELBO) and importance weighted bound (IWBO) [6] (when it is available) with 1000 samples measured in bits per pixel. For our method, we run 5 independent trials and report the averaged metric and its standard deviation. The other baselines are directly adopted from Hoogeboom et al. [19]. The result is summarized in Table 4. See Appendix A.5 for additional details.

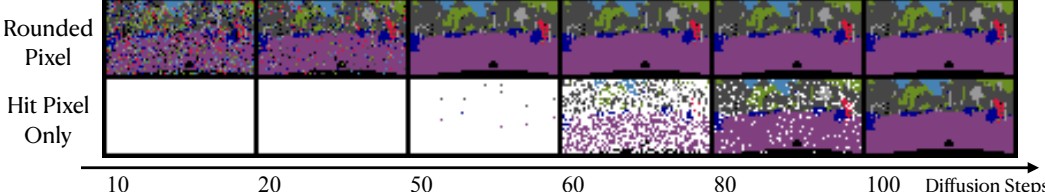

Figure 4: The trajectory of generating a segmentation map image. The upper row shows the image where the category of all pixels are decided based on the argmax (i.e., rounding) of all the 8 scores. The lower row only plots the hit pixels of the snapshots.

## 4.2 Analysis

**Hitting time distribution**   We study the hitting time distribution given by the optimized network, which is summarized in Figure 5. Our first hitting diffusion model is able to hit the domain in a well-bounded time. It is worth remarking that for Boolean and categorical distribution, *FHDM generates higher quality samples with much fewer diffusion steps.* For example, in graph generation, FHDM on average takes about 100 steps while the previous approach such as Niu et al. [30] requires 6K steps. Similarly, in segmentation map generation, FHDM takes about 90 steps on average while the multinomial diffusion [19] needs 4K steps. Decreasing the number of diffusion steps in those approaches will degenerate the performance. For example, if we only use 120 diffusion steps in Niu et al. [30] the averaged performance becomes 0.306 which is much worse. See Appendix A.6 for detailed result.

**Why we can stop at hitting time**   The key feature of FHDM that is we stop the diffusion when it hits the domain rather than keep it running for a pre-fixed time. We explore more the intuition behind such a process. In figure 4 we visualize the trajectory of generating a segmentation map. By looking at the image snapshot in the upper row where the value of each pixel is decided by the argmax (i.e. rounding) of the 8 scores, we observe that the global contour of the image is already determined at a very early time (i.e., step 20) while the refinement of local details is almost finished at step 50. Our first hitting model exploits such property to stop the diffusion of the hit pixels that the model has enough confidence about its value making the generating process of the rest pixels easier.

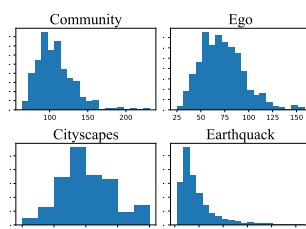

Figure 5: Hitting time distributions for different data distributions.

## 5   Conclusion

We propose the first hitting diffusion model (FHDM), which generalizes the fixed-time diffusion process and allows instance-dependent adaptive diffusion steps. Leveraging the idea of exit distribution, FHDM provides an unified framework for learning distribution in various special domains. Despite the good functionality, FHDM takes slightly larger training overhead, which is partially solved by the our fast sampling tricks.

## 6   Acknowledge

The research was conducted in both the statistical learning and AI group (SLAI) led by Qiang Liu at UT Austin. SLAI research is supported in part by CAREER-1846421, SenSE-2037267, EAGER-2041327, and Office of Navy Research, and NSF AI Institute for Foundations of Machine Learning (IFML).

## Societal Impacts

The proposed first hitting diffusion model is able to generate data in various of domains and thus might be maliciously used to generate fake data such as images or videos, which might cause negative

societal impact or even crime. Unfortunately, this paper is not able to provide technique to prevent such abuse as it is not the main focus of the paper. The theoretical foundation itself does not cause negative societal impacts, to the best of our knowledge.

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
