# A Appendix

## A.1 Algorithm for Learning Categorical Data

The over all training algorithm for learning categorical generative models is similar to the other cases. To simulate the conditioned process, given any exit point $x \in C_{d,m} \subseteq B_d$, we know that $\mathbb{Q}^{C_{d,m}}(\cdot \mid Z_\tau = x) = \mathbb{Q}^{B_d}(\cdot \mid Z_\tau = x)$ and thus (5) can be reused. The training of the network is also similar, the only difference is that we have the additional term $\nabla_z \log \text{Ber}(\Omega \mid Z_t)$ in the output of the network to ensures that the generative process in proposition 2.10 is guaranteed to hit $\Omega$ when it exists $V$. The training loss is thus

$$\mathcal{L}(\theta) = \frac{1}{2}\mathbb{E}_{\substack{x \sim \Pi^* \\ Z \sim \mathbb{Q}^x}}\Big[\int_0^\tau \big\|\mathbb{I}\{Z_t \in C_{d,m}\} \circ \big(f_t^\theta(Z_t) + \nabla_{Z_t} \log \text{Ber}(\Omega \mid Z) - \nabla_{Z_t} \log \text{Ber}(Z_t \mid x)\big)\big\|^2 dt$$
$$- \log p_0^\theta(Z_0)\Big] + const.$$

## A.2 Practical Algorithm

We give a detailed practical algorithm.

**Discretized process** Suppose that the diffusion step size at step $k$ is $\epsilon_k$. Given the exit point $x$, the discretized conditioned process can be simulated by

$$Z_{t_{k+1}} = \Big(b_{t_k}(Z_{t_k}) + \sigma_{t_k}^2(Z_{t_k})\nabla_z \log h_{t_k}^{\Pi^*}(Z_{t_k})\Big)\epsilon_k + \sqrt{\epsilon_k}\sigma_{t_k}(Z_{t_k})\xi_k, \quad Z_{t_0} \sim \mathbb{Q}_0^{\Pi^*}, \quad (16)$$

where $\xi_k \sim \mathcal{N}(0, I)$ is a standard Gaussian noise. Note that (16) is terminated at $t_k$ when $Z_{t_k}$ firstly hits the desired domain. Alternatively, we can first sample (discretized) unconditioned process by

$$Z_{t_{k+1}} = b_{t_k}(Z_{t_k})\epsilon_k + \sqrt{\epsilon_k}\sigma_{t_k}(Z_{t_k})\xi_k, \quad Z_{t_0} \sim \mathbb{Q}_0^{\Pi^*}, \quad (17)$$

And then apply the rotation operators defined in Section 2.3.2 such that the sampled trajectory ends at $x$.

**A simplified loss** Similar to Song and Ermon [41], Ho et al. [18], we use a stochastic version of loss (15), in which we only uniformly sample temporal snapshots to compute the loss.

$$\hat{\mathcal{L}}(\theta) = \frac{1}{2}\mathbb{E}_{\mathbb{Q}^{\Pi^*}}\mathbb{E}_{t \sim \text{Unif}\{0,\ldots,\tau\}}\Big[\big\|\sigma_t(Z_t)^{-1}(s_t^\theta(Z_t) - b_t(Z_t \mid Z_\tau))\big\|^2 - \log p_0^\theta(Z_0)\Big] + const. \quad (18)$$

In Algorithm A.2, we summarize the training procedure of FHDM.

---

**Algorithm 2** Learning Generative Models by First Hitting Diffusion

---

**Inputs**: A data $\{x^{(i)}\}$ drawn from $\Pi^*$ on $\Omega$. A baseline process $\mathbb{Q}$ and a model $\mathbb{P}^\theta$ that are absorbing to $\Omega$.
**Goal**: Find $\theta$ such that $\mathbb{P}_\Omega^\theta \approx \Pi^*$.
**Training**: By minimizing $\mathcal{L}(\theta)$.
(Optional) Pre-simulate unconditioned trajectories of $\mathbb{Q}$ using (17).
**for** training iters **do**
    Get a mini batch of data from training set.
    //Optionally, we can use fast bridge sampling tricks to get conditioned sample by rotating //
    pre-simulated unconditioned trajectories.
    Sample trajectories $\mathbb{Q}(\cdot \mid Z_\tau = x)$ for each data $x$ in the mini batch using (16)
    Calculate the mini-batch loss $\mathcal{L}(\theta)$ defined in Equ (18).
    Apply gradient descent to update $\theta$.
**end for**

---

## A.3 Sampling with first hitting $h$-transform

The $h$-transform formula on first hitting diffusion readily provides a simple mechanism for approximate sampling from $\Pi^*$: Assume the baseline process $X$ is designed simple enough such that the

conditional harmonic measure $\mathbb{Q}_\Omega(\cdot \mid Z_t = z)$ is easy to calculate, then we can approximately $h_t^{\Pi^*}(z)$ in (10) by Monte Carlo sampling from $\mathbb{Q}_\Omega(\cdot \mid Z_t = z)$:

$$h_t^{\Pi^*}(z) \approx \frac{1}{m} \sum_{i=1}^{m} \pi^*(x^{(i)}), \;\; x^{(i)} \sim \mathbb{Q}_\Omega(\cdot \mid Z_t = z),$$

use it simulate process (9). The gradient $\nabla \log h_t^{\Pi^*}$ can be approximated with either the reparameterization method or score function method. See Algorithm A.3.

---

**Algorithm 3** Approximate Sampling by First Hitting Diffusion

---

**Goal**: Draw sample from $\Pi^*$ on $\Omega \in \mathbb{R}^d$.

**Prepare** a baseline diffusion process $Z \sim \mathrm{Ito}_\Omega(b, \sigma)$ in (3) with exit distribution $\mathbb{Q}_\Omega(A \mid Z_t = z) = \mathbb{Q}(Z_\tau \in A \mid Z_t = z)$. Let $h = \mathrm{d}\Pi^*/\mathrm{d}\mathbb{Q}_\Omega(\cdot|Z_0 = z_0)$ be the density ratio between $\Pi^*$ and $\mathbb{Q}_\Omega(\cdot|Z_0 = z_0)$, where the initialization $Z_0 = z_0$ is in $V \setminus \Omega$.

**Simulate** the following process $\{\hat{Z}_t\}$ starting from $\hat{Z}_0 = z_0$ and stop at the first hitting time $\tau = \inf\{t \geq 0 \colon \hat{Z}_t \in \Omega\}$:

$$\mathrm{d}\hat{Z}_t = \left(b_t(\hat{Z}_t) + \sigma_t^2(\hat{Z}_t)\nabla_z \log \hat{h}_t(Z_t)\right)\mathrm{d}t + \sigma_t(\hat{Z}_t)\mathrm{d}W_t, \tag{19}$$

where $\hat{h}_t(z) = \frac{1}{m}\sum_{i=1}^{m} \pi^*(x^{(i)})$, where $\{x^{(i)}\}_{i=1}^m$ is drawn i.i.d. from $\mathbb{Q}_\Omega(\cdot \mid Z_t = z)$; the derivative $\nabla_z \log \hat{h}_t(z)$ can be calculated by either the reparameterization trick or score function method.

**Return** $\hat{Z}_\tau$ as an approximate draw from $\Pi^*$.

---

## A.4 Connection with SMLD and DDPM

Standard diffusion generative models such as SMLD and DDPM determinates the diffusion process at a fixed time, which can be included as a special first hitting model as shown in Example 2.4. We clarify the connection to SMLD and DDPM for completeness here. In this case, we set $\mathbb{Q}$ to be an Ornstein-Uhlenbeck (O-U) process $\mathrm{d}Z_t = \alpha_t Z_t \mathrm{d}t + \sigma_t \mathrm{d}W_t$ initialized at $Z_0 \sim \mathcal{N}(\mu_0, v_0)$ and stopped at a deterministic time $t = t$, where $\alpha_t \in \mathbb{R}$ and $\sigma_t \geq 0$, $v_0 \geq 0$, $\forall t$. This is a Gaussian process. Let $Z_t \sim \mathcal{N}(\mu_t, v_t)$. Denote by $\bar{Z}_t = Z_{t-t}$ the time reversed process, which follows [2]

$$\mathrm{d}\bar{Z}_t = \left(-\alpha_{t-t}\bar{Z}_t + \sigma_{t-t}^2 \frac{\mu_{t-t} - \bar{Z}_t}{v_{t-t}}\right)\mathrm{d}t + \sigma_{t-t}\mathrm{d}\bar{W}_t,$$

where $\bar{W}_t$ is a copy of standard Brownian motion. If we set $v_0 \to +\infty$ in the initial $Z_0$, we expect to have $v_t \to +\infty$ under proper regularity conditions on $\alpha_t$ and $\sigma_t$, the second term in the drift of $\bar{Z}_t$ is canceled, yielding $\mathrm{d}\bar{Z}_t = -\alpha_{t-t}\bar{Z}_t \mathrm{d}t + \sigma_{t-t}\mathrm{d}\bar{W}_t$. This then reduces to the processes used in SMLD ($\alpha_t = 0$), and DDPM and SDE method in [43] ($\alpha_t > 0$). This framework of learning fixed-time diffusion models using bridge processes are explored separately in a recent work [34]. The authors devote more in-depth discussions on the fixed-time diffusion case in a separate work.

## A.5 Additional Experiment Details

### A.5.1 Point Cloud Generation

**Training details** The ShapeNet dataset contains 51,127 shapes from 55 categories and is randomly split training, testing and validation set by the ratio 80%, 15% and 5%. For each shape, we sample 2048 points to acquire the point clouds and normalize each of them to zero mean and unit variance.

We build our method on Luo and Hu [26] in which the encoder of a flow-based model is used to learn a latent code of the shape and conditioning on the shape latent code, the point are independently generated based on a diffusion model. We substitute the DDPM-type [18] of diffusion model with ours and all the other components remain the same. Each point is generated using 100 diffusion steps and the step size linearly decays starting from 0.02 to $10^{-4}$. We use the same network architecture for flow-based model and point diffusion network. We train the model for 1M steps with batch size 128 using Adam optimizer [21].

**More Details on Evaluation Metrics** Both MMD, JSD and 1-NN measures the fidelity of the generated samples. The 1-NN score is the accuracy of 1-NN classifier in predicting whether a point cloud is generated by the model or from the data. Lower 1-NN scores suggest higher quality. MMD and JSD measures the probability distance between the point distributions of the generated set and the reference set from data and thus lower MMD and JSD means higher quality. COV detects mode-collapse and higher COV suggests more diverse generated samples.

### A.5.2 Generating Distribution on Sphere

**Training details** All datasets are split into training, validation and test sets with $(0.8, 0.1, 0.1)$ proportions. We train the model for 2000 iterations using Adam Optimizer [21] with learning rate 0.05 and batch size 128. We use a three-layer MLP with 100 hidden units and ReLU activation to approximate the drift. We set the maximum diffusion step as 10K with step size $5 \times 10^{-4}$. The model on average takes 1K steps to hit and seldom takes more than 5K to hit. See section 4.2 for more details on the hitting time distribution.

### A.5.3 Graph Generation

**Training details** We set the maximum number of SDE steps as 10K, and it takes on average about 100 steps to hit. See section 4.2 for more detailed analysis. At each step, we set the standard deviation of gaussian noise as 0.5. We initialize all the coordinate 0.5 and stop the updating of a coordinate at the first time its distance to 0 or 1 is less than 0.05. We use the same network architecture and training pipeline as Niu et al. [30]. Adam optimizer [21] with 0.001 learning rate is applied. Batch size is set to 32 and for each graph, we randomly sample 6 snapshot in the trajectory for training. The score matching loss of a hit coordinate is masked out at training.

### A.5.4 Segmentation Map Generation

**Training details** Our network architecture and training pipeline is almost the same as the multinomial diffusion model proposed in Hoogeboom et al. [19]. The only architecture difference is that Hoogeboom et al. [19] first feed the image into an embedding layer before passing to the subquential U-Net [35] like structure while we use a linear layer with the same output dimension. This is because the multinomial diffusion model [19] is a discrete diffusion in which the value at each pixel is considered to be discrete while FHDM is a continuous diffusion. We set the number of maximum diffusion steps to be 100 and the step size to be 0.1. We apply step-decayed Gaussian noise at different diffusion steps, in which the standard deviation at initial is 1 and decay to half at step 500 and 750. The pixel is hit and stopped to update at the first time its largest categorical score (among 8 of them) is greater than $1 - \epsilon$ with $\epsilon = 0.01$. We apply the same data augmentation and train the model for 500 epochs with batch size 64, learning rate $10^{-4}$ and Adam optimizer [21]. For each image in the batch, we randomly sample one time snapshot along the diffusion trajectory for training.

For this task, we apply the fast bridge sampling method proposed in Section 2.3.2. At the beginning of each epoch, we generate $10 \times$ batch size $\times H \times W$ unconditional SDE trajectories where $H, W$ is the height and width of the images. At the training time, to simulate the SDE trajectories of a given image in the training set, for each pixel, we randomly select one saved unconditional SDE trajectories and rotates it such that it ends at that pixel.

### A.6 Additional Experiment Results

**Number of diffusion steps** When we restrict the number of diffusion steps of EDP-GNN [30], the second-best approach, similar to that of FHDM (120), we observe a significant performance drop. As shown in Table 5, the performance of EDP-GNN degenerates badly when we decrease its diffusion steps from 4K to 120.

**Acceleration by fast sampling** We give brief analysis on the acceleration effect of the fast bridge sampling method described in Section 2.3.2. When applied to the segmentation generation experiment, we pre-simulate 640 trajectories in the beginning of each training epoch which gives 2.5x acceleration from 24.3 cpu time/epoch to 9.8 cpu time/epoch, making the training time of FHDM is comparable to Hoogeboom et al. [19] (6.5 cpu time/epoch). We remark that although FHDM has slightly larger training overhead, its only requires less than 100 diffusion steps at inference, giving a 40x speed up compared with Hoogeboom et al. [19].

| Method | Community-small | | | | Ego-small | | | | Avg |
|---|---|---|---|---|---|---|---|---|---|
| | Deg. | Clus. | Orbit. | Avg. | Deg. | Clus. | Orbit. | Avg. | |
| EDP-GNN | 0.053 | 0.144 | 0.026 | 0.074 | 0.052 | 0.093 | 0.007 | 0.050 | 0.062 |
| EDP-GNN (step=120) | 0.586 | 0.253 | 0.705 | 0.515 | 0.141 | 0.114 | 0.036 | 0.097 | 0.306 |
| Ours | **0.004** | **0.104** | **0.001** | **0.036** | **0.019** | **0.047** | **0.005** | **0.024** | **0.030** |

Table 5: Comparing FHDM with EDP-GNN with similar diffusion steps.

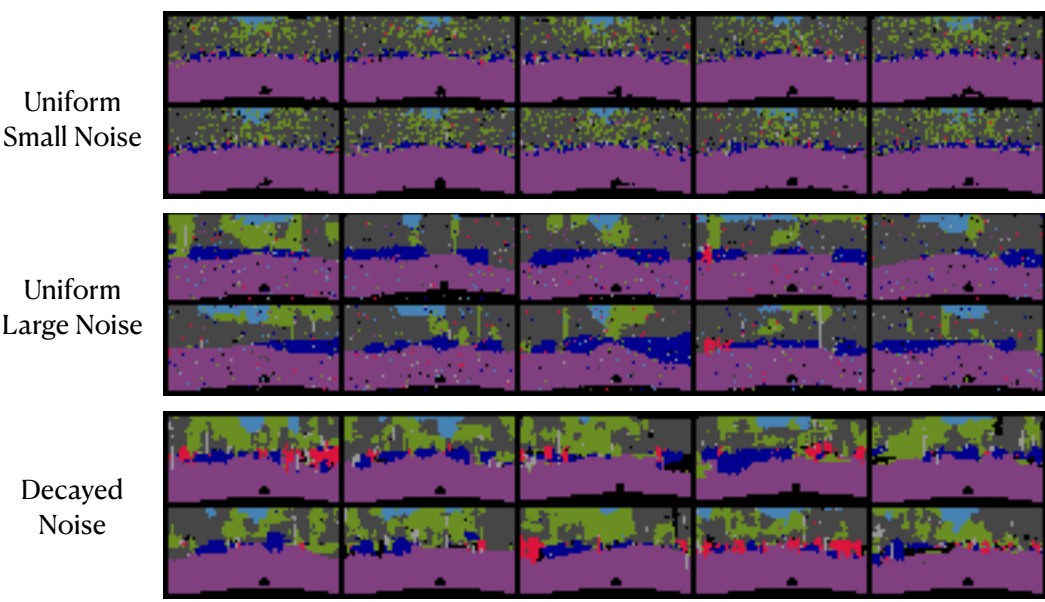

Figure 6: Compare the generated segmentation maps with different noise schedule.

**Ablation studies on noise schedule** In practice, we observe that the design of the noise schedule can be important for some tasks such as segmentation generation. We show samples generated by FHDM with uniformly small noise (std=0.25), uniformly large noise (std=1) and decayed noise as described in Section in 4.1. It is worth noticing that using a uniformly small noise generates over-smoothed and degenerated images that fail to reveal the details while using a uniformly large noise gives more diverse but noisy images. In comparison, the decaying noise generates high-quality diverse images with fine details.

## A.7 Visualization of Generated Samples

**Point Cloud Generation** Please see Figure 7 and 8 for the generated airplane and chair point cloud using FHDM.

**Generating Distribution on Sphere** Please see Figure 11 for the generated graphs using FHDM.

**Segmentation Map Generation** Please see Figure 9 for the generated distributions on sphere by FHDM.

**Graph Generation** Please see Figure 11 for the generated graphs using FHDM.

**Segmentation Map Generation** Please see Figure 10 for the generated segmentation maps by FHDM.

## A.8 Discretization Error Analysis

Consider the following Ito process

$$dZ_t = b(Z_t)dt + \sigma(Z_t)dW_t$$

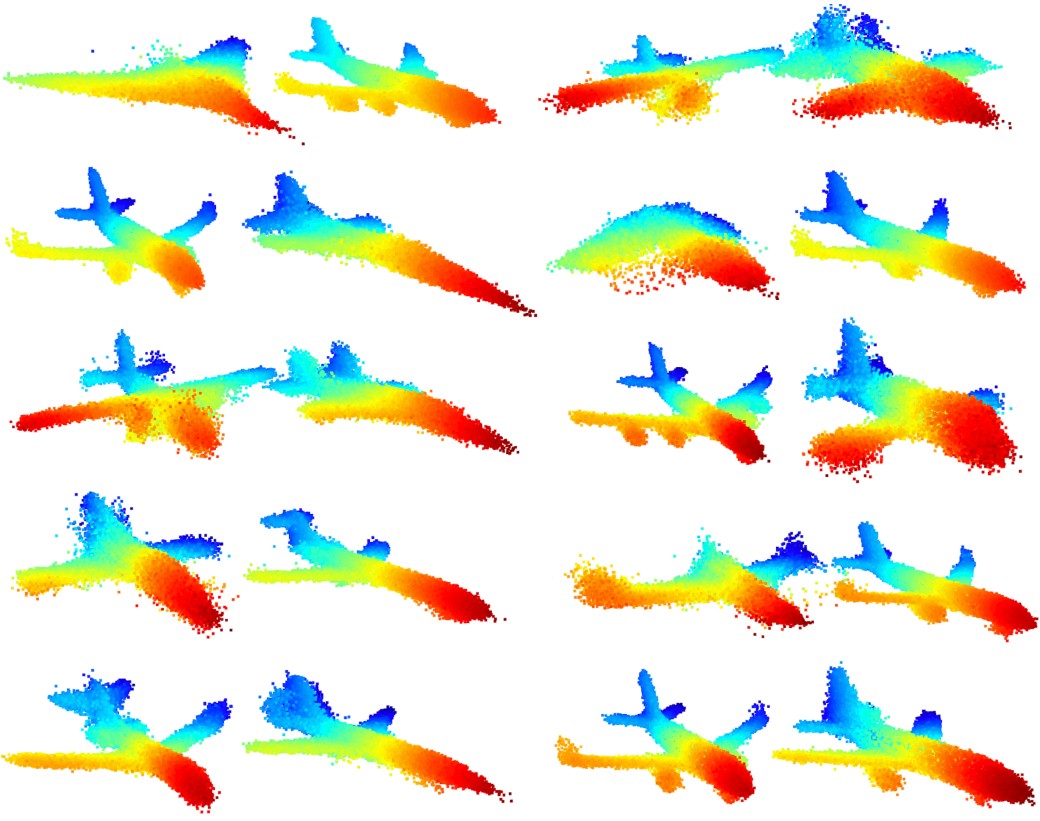

Figure 7: The generated airplane point cloud by FHDM.

and a open subset $V \subseteq \mathbb{R}^d$. Here $b \in \mathbb{R}^d$ is the drift and $\sigma \in \mathbb{R}^{d \times d}$ is the diffusion matrix. We stop the process when $Z_t$ hit the domain $\Omega$ and denote the hitting time as $\tau := \inf_{t \geq 0}\{Z_t \notin V\}$. We consider the discretalization error of the conditional distribution $\pi_T$ with temporal truncation $T$, i.e,

$$\pi_T := \text{law of } X_\tau \mid \tau \leq T.$$

This corresponds to the situation that we discard the non-hit process after waiting for $T$ time. To simulate the above process, we consider the Euler discretalization on $[0, T]$. Suppose $R$ is a set of grid points on $[0, T]$ in which we define

$$r_t = \max\{r \in R : r \leq t\}.$$

The Euler discretized process is thus defined as

$$d\bar{Z}_t = b(\bar{Z}_{r_t}) + \sigma(\bar{Z}_{r_t})dW_t.$$

And similarly, we can define its (discretize) stopping time $\bar{\tau} := \min_{r \in R}\{r : \bar{Z}_{r_t} \notin V\}$. We want to bound the discrepancy between $\pi_T$ and the following distribution

$$\bar{\pi}_T := \text{law of } \bar{X}_{\bar{\tau}} \mid \bar{\tau} \leq T.$$

We consider the Wassestein distance $\mathcal{W}[\bar{\pi}_T, \pi_T]$ for measuring the discrepancy. In this section, $|| \cdot ||$ is vector norm when applied to vector and is matrix operator norm when applied to matrix.

**Assumption 1.** $b$ and $\sigma$ is $L$-Lipschitz and $\sup_z(||b(z)|| + ||\sigma(z)||) \leq L$.

**Assumption 2.** There exists a bounded $C_b^2$ function $\delta : \mathbb{R}^d \to \mathbb{R}$ such that $\delta > 0$ on $V$, $\delta = 0$ on $\Omega$ and $\delta < 0$ on $\mathbb{R}^d \setminus (V \cup \Omega)$ and satisfies the non-characteristic boundary condition $||\sigma \nabla \delta|| \geq 2L^{-1}$ on $\{||\delta|| \leq r\}$ for some $r > 0$.

Assumption 1 is a standard assumption on the Lipschitz continuity and boundedness on the drift and diffusion function. Assumption 2 is more on a technical condition and is introduced in Bouchard et al. [4] and intuitively it can be understood in the way that there exists a bounded smooth function that can indicate whether we are within $V$ or out of $V$.

**Theorem 1.** *Let* $\Delta := \min_{r_t \neq r_{t'}} |r_t - r_{t'}|$. *Under Assumption 1, 2 and assume that $T$ is properly large such that $\mathbb{P}(\tau \geq T - 1) \leq 1/4$, we have, there exists $\epsilon > 0$ such that for any $\Delta \leq \epsilon$,*

$$\mathcal{W}^2[\bar{\pi}_T, \pi_T] = O(\exp(cT)\Delta),$$

*for some absolute constant $c < \infty$.*

Intuitively, we show that when $T$ is properly large (which is true in practice as we should wait the process a reasonably enough time for hitting) and the step size is small enough, the discretalize error is small.

*Proof.* Throughout the proof, $c$ denotes absolute constant and may vary in different lines. We consider the temporal augmented process $Y_t = [Z_t, t]$ in which

$$dY_t = \tilde{b}(Y_t)dt + \tilde{\sigma}(Y_t)dW_t.$$

Here $\tilde{b}$ and $\tilde{\sigma}$ are defined as

$$\tilde{b}(y) = [b(x), 1], \tilde{\sigma}_y(y) = \begin{bmatrix} \sigma(y) & \mathbf{0} \\ \mathbf{0}^\top & 0 \end{bmatrix}.$$

It is not hard to verify the Lipschitz continuity and boundedness of $\tilde{b}$ and $\tilde{\sigma}$. We also define the hitting set of the process $Y_t$ by $\tilde{V} = \{y : x \in V \text{ or } t < T + 1\}$. It is easy to show that $tildeV$ is a closed subset of $\mathbb{R}^{d+1}$ and the stopping time $\tau := \inf_{t \geq 0}\{t : Y_t \notin \tilde{V}\} \leq T + 1$. Similarly, we can define the discretized version

$$d\bar{Y}_t = \tilde{b}(\bar{Y}_{r_t})dt + \tilde{\sigma}(\bar{Y}_{r_t})dW_t.$$

Here we slightly abuse the notation of $\tau$ and $\bar{\tau}$, making them denoting the hitting time of process $Y_t$ and $\bar{Y}_t$ rather than $Z_t$ and $\bar{Z}_t$. We introduce the following Lemma used in Bouchard et al. [4].

**Lemma 1** (Theorem 3.11 in Bouchard et al. [4]). *Under assumption 1 and 2, there exists $\epsilon > 0$ such that when $\Delta \leq \epsilon$, $\mathbb{E}[|\tau - \bar{\tau}|] \leq c\Delta^{1/2}$ for some constant $c > 0$.*

Note that

$$\mathbb{E}[||\bar{Y}_{\bar{\tau}} - Y_\tau||^2 \mid \bar{\tau} \leq T] \leq \int \mathbb{E}[||\bar{Y}_{\bar{\tau}} - Y_\tau||^2 \mid |\bar{\tau} - \tau| = s, \bar{\tau} \leq T] \Pr(|\bar{\tau} - \tau| = s \mid \bar{\tau} \leq T)ds.$$

Note that we can decompose

$$\mathbb{E}[||\bar{Y}_{\bar{\tau}} - Y_\tau||^2 \mid |\bar{\tau} - \tau| = s, \bar{\tau} \leq T]$$
$$\leq 2\mathbb{E}[||\bar{Y}_{\bar{\tau}} - Y_{\bar{\tau}}||^2 \mid |\bar{\tau} - \tau| = s, \bar{\tau} \leq T] + 2\mathbb{E}[||Y_{\bar{\tau}} - Y_\tau||^2 \mid |\bar{\tau} - \tau| = s, \bar{\tau} \leq T].$$

Using Lemma A.2 in Bouchard et al. [4] and Holder's inequality, we have

$$\leq \mathbb{E}[||\bar{Y}_{\bar{\tau}} - Y_{\bar{\tau}}||^2 \mid |\bar{\tau} - \tau| = s, \bar{\tau} \leq T] \leq \sup_{t \in [0, T+s]} ||\bar{Y}_{\bar{\tau}} - Y_{\bar{\tau}}||^2 \leq c\Delta,$$

for some constant $c$. Also, by the boundedness of $\tilde{b}$

$$||Y_{\bar{\tau}} - Y_\tau|| = ||\int_{\min(\bar{\tau}, \tau)}^{\max(\bar{\tau}, \tau)} \tilde{b}(Y_t)dt|| \leq (L+1)||\tau - \bar{\tau}||.$$

Using these two bounds,

$$\mathbb{E}[||\bar{Y}_{\bar{\tau}} - Y_\tau||^2 \mid |\bar{\tau} - \tau| = s, \bar{\tau} \leq T] \leq c(\Delta + ||\tau - \bar{\tau}||^2).$$

This gives that

$$\mathbb{E}[||\bar{Y}_{\bar{\tau}} - Y_\tau||^2 \mid \bar{\tau} \leq T] \leq \int_0^T c(\Delta + ||\tau - \bar{\tau}||^2)\Pr(|\bar{\tau} - \tau| = s \mid \bar{\tau} \leq T)ds$$

$$\leq c\left(\Delta + \int_0^T s^2\Pr(|\bar{\tau} - \tau| = s \mid \bar{\tau} \leq T)ds\right).$$

Now we proceed to bound

$$\Pr(|\bar{\tau} - \tau| = s \mid \bar{\tau} \leq T) = \frac{\Pr(|\bar{\tau} - \tau| = s, \bar{\tau} \leq T)}{\Pr(\bar{\tau} \leq T)} \leq \frac{\Pr(|\bar{\tau} - \tau| = s)}{\Pr(\bar{\tau} \leq T)}.$$

Note that

$$\begin{aligned}
\Pr(\bar{\tau} > T) &= \int_0^{T+1} \Pr(\bar{\tau} > T, \tau = s) ds \\
&= \int_0^{T-1} \Pr(\bar{\tau} > T, \tau = s) ds + \int_{T-1}^{T+1} \Pr(\bar{\tau} > T, \tau = s) ds \\
&\leq (T-1) \Pr(|\bar{\tau} - \tau| \geq 1) + \int_{T-1}^{\infty} \Pr(\tau = s) ds \\
&\leq (T-1) \mathbb{E}(|\bar{\tau} - \tau|) + (1 - F_{\tau}(T - 1)) \\
&\leq c\Delta + (1 - F_{\tau}(T - 1)),
\end{aligned}$$

where $F_{\tau}$ denotes the CDF of $\tau$. When $T$ is properly large and $\Delta$ is small enough, we have $\Pr(\bar{\tau} > T) \leq 1/2$ and thus $\Pr(\bar{\tau} \leq T) = 1 - \Pr(\bar{\tau} > T) \geq 1/2$. This implies that

$$\Pr(|\bar{\tau} - \tau| = s \mid \bar{\tau} \leq T) \leq 2\Pr(|\bar{\tau} - \tau| = s).$$

We thus conclude that

$$\begin{aligned}
&\int_0^T s^2 \Pr(|\bar{\tau} - \tau| = s \mid \bar{\tau} \leq T) ds \\
&\leq 2 \int_0^T s^2 \Pr(|\bar{\tau} - \tau| = s) ds \\
&\leq 2T \int_0^T s \Pr(|\bar{\tau} - \tau| = s) ds \\
&= 2T \mathbb{E}(|\bar{\tau} - \tau|) \\
&\leq c\Delta.
\end{aligned}$$

We finally conclude that

$$\mathcal{W}^2[\bar{\pi}_T, \pi_T] \leq \mathbb{E}[||\bar{Z}_{\bar{\tau}} - Z_{\tau}||^2 \mid \bar{\tau} \leq T] \leq \mathbb{E}[||\bar{Y}_{\bar{\tau}} - Y_{\tau}||^2 \mid \bar{\tau} \leq T] \leq c\Delta.$$

$\square$

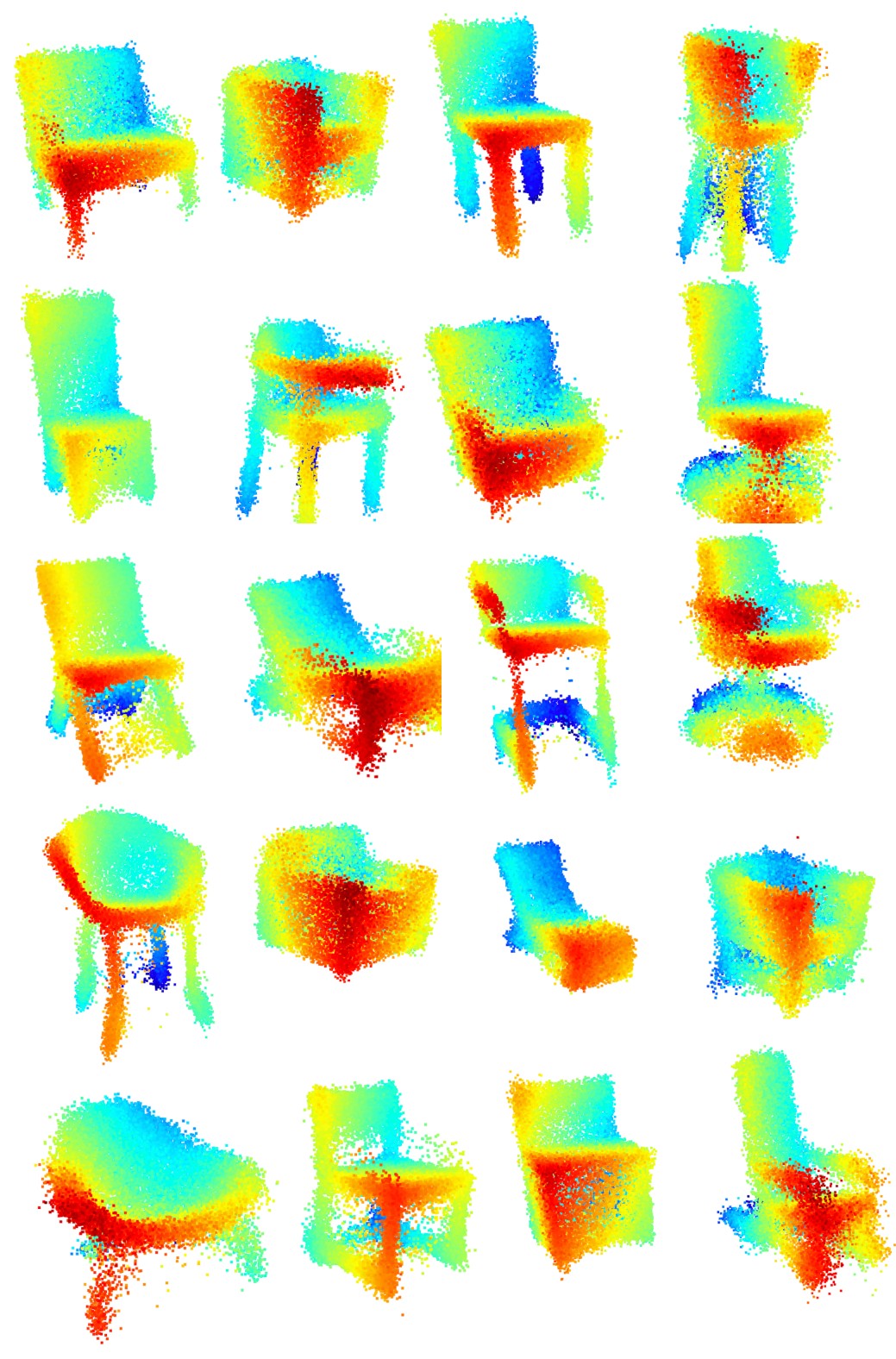

Figure 8: The generated chair point cloud by FHDM.

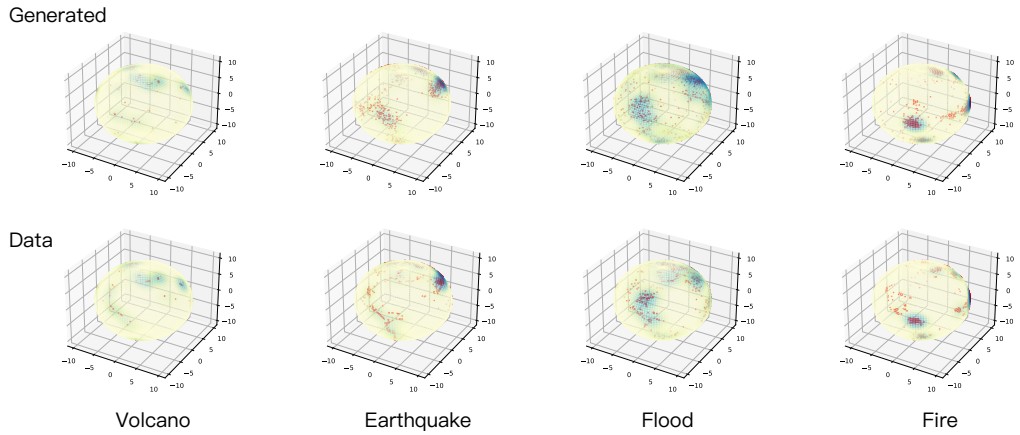

Figure 9: The generated distrubution on sphere by FHDM.

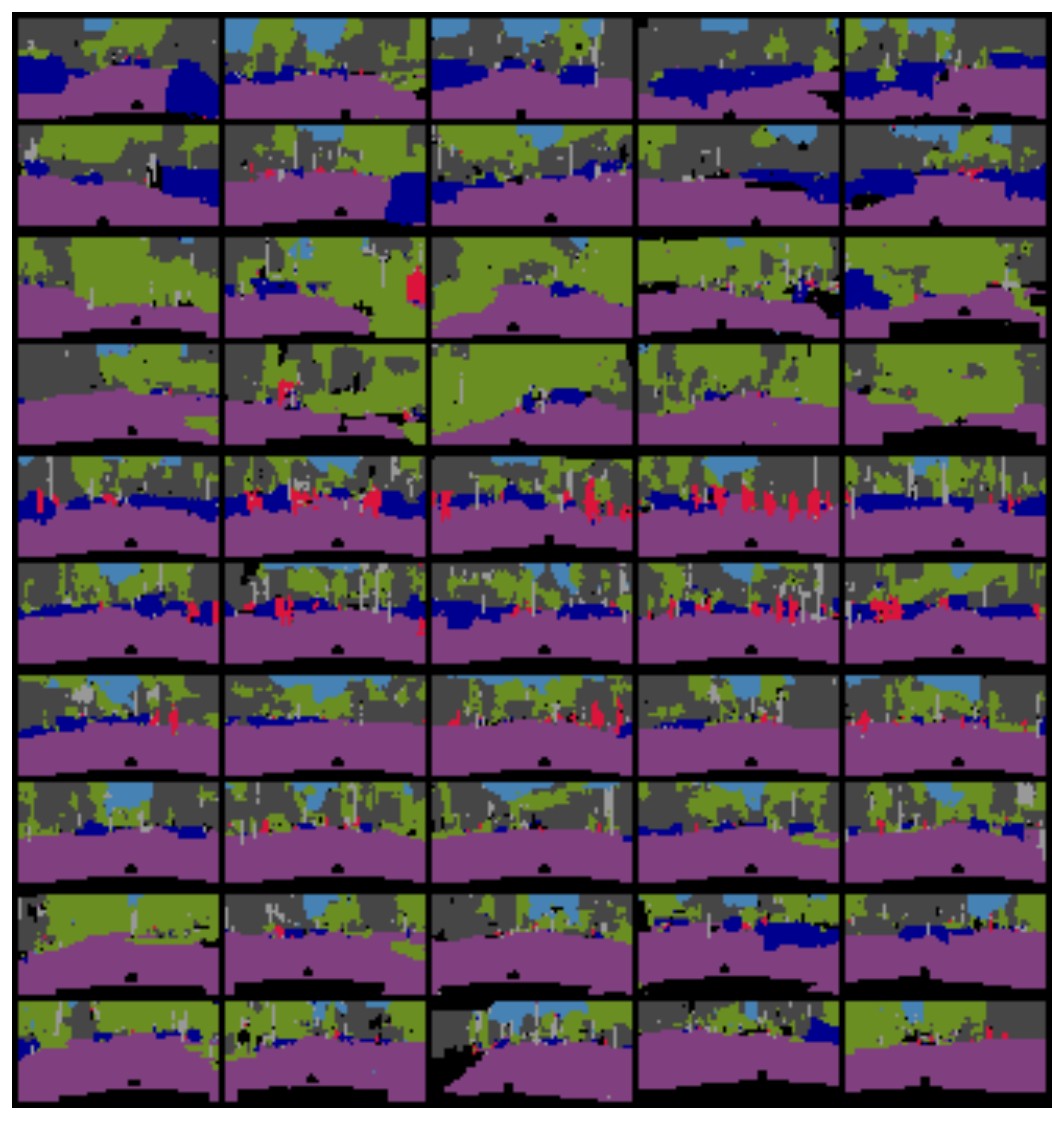

Figure 10: The generated segmentation maps by FHDM.

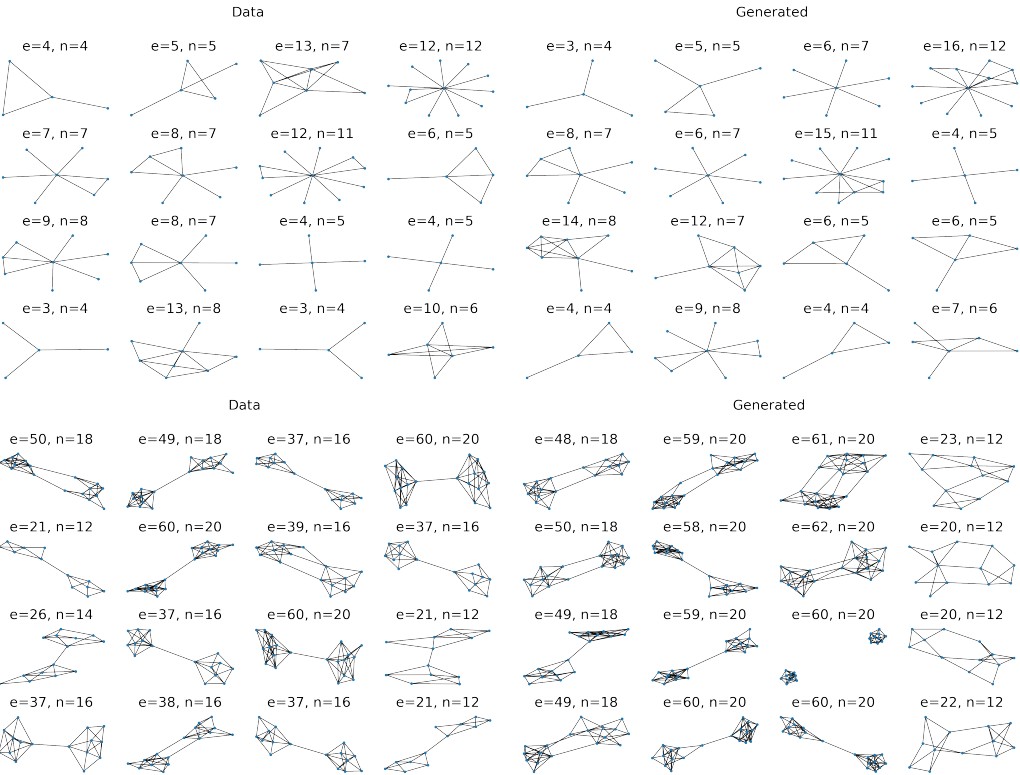

Figure 11: True and generated graphs of ego (upper rows) and community (lower rows) datasets.

## A.9 Proofs

*Proof of Proposition 2.3.* As $\mathbb{Q}^{B_d}$ is a product of identical and independent one-dimensional processes, it is sufficient to consider the one dimension case $d = 1$, in which case the process is a Brownian motion $\mathrm{d}Z_t = \mathrm{d}W_t$ starting from interval $Z_0 \in [0, 1]$ and stopped $\tau = \min_t\{t\colon Z_t \notin (0, 1)\}$ when it exits the interval. Hence, the Poisson kernel is

$$\mathbb{Q}_\Omega^{B_d}(x \mid z) = \Pr(W_\tau = x \mid W_t = z) = \Pr(W_\tau = x \mid W_t = z), \quad \forall x \in \{0, 1\}$$

Then, it is a textbook result that $\Pr(W_\tau = x \mid W_t = z) = xz + (1 - x)(1 - z) = \mathrm{Ber}(x|z)$. See e.g., Eq. 3.0.4, Page 212 of Borodin and Salminen [3]. $\qquad\square$

*Proof of Proposition 2.5.* It is a straightforward application of formula (4) in the case of $b_t = 0$, $\sigma_t(Z_t) = \mathbb{I}(\|Z_t\| < 1)$ and $q_\Omega(x|z) = \frac{1-\|z\|^2}{\|x-z\|^d}$ as shown in (2). $\qquad\square$

*Proof of Proposition 2.6.* It is a straightforward application of formula (4) in the case of $b_t = 0$, $\sigma_t(Z_t) = \mathrm{diag}(\mathbb{I}(Z_t \in (0, 1)))$ and $\mathbb{Q}_\Omega(x|z) = \mathrm{Ber}(x|z)$ as shown in Proposition (2.3). $\qquad\square$

*Proof of Proposition 2.7.* This is the standard result on Brownian bridge. In particular, we just need to note that $(Z_T \mid Z_t = z) \sim \mathcal{N}(z, T - t)$ and apply formula (4). $\qquad\square$

*Proof of Proposition 2.8.* Eq. (13) is the direct result of $\mathbb{Q}^{\Pi^*} = \arg\min_{\mathbb{P}} \mathcal{KL}(\mathbb{P} \,||\, \mathbb{Q}^{\Pi^*})$, and that

$$\mathcal{KL}(\mathbb{P} \,||\, \mathbb{Q}^{\Pi^*}) \equiv \mathcal{KL}(\mathbb{P} \,||\, \mathbb{Q}) - \mathbb{E}_{\mathbb{P}}[\log \pi^*(Z_\tau)],$$

where we used $\mathbb{Q}^{\Pi^*}(\mathrm{d}Z) = \mathbb{Q}(\mathrm{d}Z)\pi^*(Z_\tau)$.

Eq (12) is a simple consequence of the disintegration theorem. Note that any $\mathbb{P}$ that satisfies $\mathbb{P}_\Omega = \Pi^*$ can be written into $\mathbb{P}(\mathrm{d}Z) = \Pi^*(\mathrm{d}Z_\tau)\mathbb{P}(\mathrm{d}Z \mid Z_\tau)$. By the chain rule of KL divergence,

$$\mathcal{KL}(\mathbb{P} \,||\, \mathbb{Q}) = \mathcal{KL}(\Pi^* \,||\, \mathbb{Q}_\Omega) + \mathbb{E}_{Z_\tau \sim \Pi^*}\left[\mathcal{KL}(\mathbb{P}(\cdot|Z_\tau) \,||\, \mathbb{Q}(\cdot|Z_\tau))\right]. \tag{20}$$

Since it is constrained that $\mathbb{P}_\Omega = \Pi^*$, the optimal $\mathbb{P}$ is determined by the choice of $\mathbb{P}(\cdot|Z_\tau)$ and it should yield $\mathbb{P}(\mathrm{d}Z \mid Z_\tau) = \mathbb{Q}(\mathrm{d}Z \mid Z_\tau)$. Therefore, the optimal $\mathbb{P}$ is $\Pi^*(Z_\tau)\mathbb{Q}(\cdot|Z_\tau) = \mathbb{Q}^{\Pi^*}$.

In fact, by the same derivation, we can see that (12) remains correct if we replace $\mathcal{KL}(\mathbb{P} \,||\, \mathbb{Q})$ with $\mathcal{KL}(\mathbb{Q} \,||\, \mathbb{P})$. $\qquad\square$

*Proof of Proposition 2.9.* Denote by $\mathbb{Q}^x = \mathbb{Q}(\cdot|Z_\tau = x)$. Let $p^\theta$ be the density function of $\mathbb{P}^\theta$ w.r.t. to some reference measure (e.g., $\mathbb{Q}^{\Pi^*}$). We have

$$\mathcal{KL}(\mathbb{Q}^{\Pi^*} \,||\, \mathbb{P}^\theta) = -\mathbb{E}_{Z \sim \mathbb{Q}^{\Pi^*}}[\log p^\theta(Z)] + const$$
$$= -\mathbb{E}_{x \sim \Pi^*}\mathbb{E}_{Z \sim \mathbb{Q}^x}[\log p^\theta(Z)] + const$$
$$= \mathbb{E}_{x \sim \Pi^*}\left[\mathcal{KL}(\mathbb{Q}^x \,||\, \mathbb{P}^\theta)\right] + const,$$

where $\mathcal{KL}(\mathbb{Q}^x \,||\, \mathbb{P}^\theta)$ can be evaluated using Girsanov theorem,

$$\mathcal{KL}(\mathbb{Q}^x \,||\, \mathbb{P}^\theta) = \mathcal{KL}(\mathbb{Q}_0^x \,||\, \mathbb{P}_0^\theta) + \frac{1}{2}\mathbb{E}_{Z \sim \mathbb{Q}^x}\left[\int_0^\tau \left\|s_t^\theta(Z_t) - b_t(Z_t|x)\right\|_2^2 \mathrm{d}t\right]$$
$$= \mathbb{E}_{Z \sim \mathbb{Q}^x}\left[-\log p_0^\theta(Z_0) + \frac{1}{2}\int_0^\tau \left\|s_t^\theta(Z_t) - b_t(Z_t|x)\right\|_2^2 \mathrm{d}t\right] + const.$$

Hence

$$\mathcal{L}(\theta) = \mathbb{E}_{x \sim \Pi^*, Z \sim \mathbb{Q}^x}\left[-\log p_0^\theta(Z_0) + \frac{1}{2}\int_0^\tau \left\|s_t^\theta(Z_t) - b_t(Z_t|x)\right\|_2^2 \mathrm{d}t\right] + const$$
$$= \mathbb{E}_{Z \sim \mathbb{Q}^{\Pi^*}}\left[-\log p_0^\theta(Z_0) + \frac{1}{2}\int_0^\tau \left\|s_t^\theta(Z_t) - b_t(Z_t|Z_\tau)\right\|_2^2 \mathrm{d}t\right] + const.$$

$\qquad\square$

*Proof of Proposition 2.10.* Assume $\|f\|_\infty := \sup_{t\in[0+\infty),x\in[0,1]^d}\|f_t(x)\|_2 < +\infty$. Consider the following two processes with the same initialization:

$$
\begin{aligned}
\mathbb{Q}^0\colon\ & dZ_t = \mathbb{I}(Z_t \in (0,1)) \circ (\nabla_z \log \mathrm{Ber}(\Omega \mid Z_t)dt + dW_t) \\
\mathbb{Q}^f\colon\ & dZ_t = \mathbb{I}(Z_t \in (0,1)) \circ \left(f_t^\theta(Z_t) \,+\, \nabla_z \log \mathrm{Ber}(\Omega \mid Z_t)dt + dW_t\right).
\end{aligned}
\tag{21}
$$

Girsanov theorem shows that $\mathcal{KL}(\mathbb{Q}^0 \,\|\, \mathbb{Q}^f) = \frac{1}{2}\mathbb{E}_{\mathbb{Q}^0}[\int_0^\tau \|f_t(Z_t)\|^2] \leq \frac{1}{2}\|f\|_\infty^2 \mathbb{E}_{\mathbb{Q}^0}[\tau] < +\infty$, where we used the fact that the expected hitting time $\mathbb{E}_{\mathbb{Q}^0}[\tau]$ of $\mathbb{Q}^0$ is finite (see Lemma A.1 below).

Now $\mathcal{KL}(\mathbb{Q}^0 \,\|\, \mathbb{Q}^f) < +\infty$ implies that that $\mathbb{Q}^0$ and $\mathbb{Q}^f$ has the same support. Hence the fact that $\mathbb{Q}^0$ guarantees to hit $\Omega$ when exiting $V$, i.e., $\mathbb{Q}^0(Z_\tau \in \Omega) = 1$, when exit implies that $\mathbb{Q}^f$ has the same property, i.e., $\mathbb{Q}^f(Z_\tau \in \Omega) = 1$.

**Lemma A.1.** *Let* $\tau^0 = \inf\{t\colon Z_t \in \Omega\}$ *be the first hitting time to* $\Omega \subseteq \{0,1\}^d$ *of the the process* $\mathbb{Q}^0$ *in Eq. (21). Then* $\mathbb{E}[\tau^0] < +\infty$.

*Proof.* Consider the following two processes starting from the same deterministic initialization $Z_0 = Y_0 = z_0 \in (0,1)^d$:

$$
\begin{aligned}
\mathbb{Q}^0\colon\ & dZ_t = \mathbb{I}(Z_t \in (0,1)) \circ (\nabla_z \log \mathrm{Ber}(\Omega \mid Z_t)dt + dW_t) \\
\mathbb{Q}^*\colon\ & dY_t = \mathbb{I}(Y_t \in (0,1)) \circ (dW_t).
\end{aligned}
$$

Denote by $\tau^0$ and $\tau^*$ the corresponding hitting times to $\Omega$, that is, $\tau^0 = \inf\{t\colon Z_t \in \Omega\}$, and $\tau^* = \inf\{t\colon Y_t \in \Omega\}$.

Then we know from $h$-transform that $\mathbb{Q}^0$ is the conditioned process of $\mathbb{Q}^*$ given that $Z_\tau \in \Omega$, that is, $\mathbb{Q}^0 = \mathbb{Q}^*(\cdot \mid Z_\tau \in \Omega)$.

Therefore, the first hitting time $\tau^0$ of $\mathbb{Q}^0$ has the same law as that of $\tau^* \mid Y_\tau \in \Omega$, that is, $\mathbb{Q}^0(\tau^0 \in A) = \mathbb{Q}^*(\tau^* \in A \mid Y_\tau \in \Omega)$ for any measurable set $A \subseteq [0, +\infty)$.

But we know that $\mathbb{E}[\tau^* \mid Y_\tau \in \Omega] < +\infty$ due to the diffusion nature of Brownian motion. Hence, $\mathbb{E}[\tau^0] = \mathbb{E}[\tau^* \mid Y_\tau \in \Omega] < +\infty$. $\qquad\square$

$\qquad\square$

## A.10  More Discussions on First Hitting Diffusion Models on $\mathbb{R}^d$

Assume the distribution $\Pi^*$ of interest is on $\mathbb{R}^d$. To design first hitting diffusion models that yield results on $\Pi^*$, we embed $\mathbb{R}^d$ into the hyperplane $\Omega := \{(x,y) \in \mathbb{R}^{d+1}\colon y = y_{\max}\}$ in $\mathbb{R}^{d+1}$ where $y_{\max}$ is a constant (e.g., $y_{\max} = 1$). We construct a baseline process $\bar{\mathbb{Q}}$ to be a diffusion process on $\mathbb{R}^{d+1}$:

$$
\bar{\mathbb{Q}}\colon\quad dZ_t = dW_t, \quad dY_t = b(Y_t,t)dt + \sigma d\tilde{W}_t, \quad Z_0 = z_0 \in \mathbb{R}^d,\ Y_0 = 0,
\tag{22}
$$

where $W_t$ and $\tilde{W}_t$ are independent Brownian motions in $\mathbb{R}^d$ and $\mathbb{R}$, respectively.

We can think $Y_t$ as an "effective age" of the particle $Z_t$, and the sample is collected when $Y_t = y_{\max}$. Therefore, the hitting time of interest is $\tau := \{t\colon (X_t, Y_t) \in \Omega\} = \{t\colon Y_t = y_{\max}\}$.

A special case is $\sigma = 0$ and $b(Y_t, t) = 1$, in which case $(Z_t, Y_t)$ hits the target domain $\Omega$ in the fixed time $t = y_{\max}$. This corresponds to the standard denoising diffusion models [e.g., 43].

Another extreme case is to take $b = 0$, so that $Y_t$ is a Brownian motion without a drift. In this case, the hitting time follows an inverse Gamma distribution, and the exit distribution is a Cauchy distribution:

$$
(\tau \mid Z_t, Y_t) \sim \mathrm{InvGamma}\left(\frac{1}{2}, \frac{(y_{\max} - Y_t)^2}{2\sigma^2}\right), \quad (Z_\tau \mid Z_t, Y_t) \sim \mathrm{Cauchy}\left(Z_t, \frac{y_{\max} - Y_t}{2}\right),
$$

where the density of $\mathrm{InvGamma}(\alpha, \beta)$ is $f(x; \alpha, \beta) = \frac{\beta^\alpha}{\Gamma(\alpha)} x^{-(\alpha+1)} \exp(-\beta/x)$, and density of $\mathrm{Cauchy}(\mu, s)$ is $f(x; \mu, s) \propto (s^2 + \|x - \mu\|^2)^{-(d+1)/2}$.

An advantage of using random hitting is that it allows us to spend less time on generating $Z_t$ that is close to the starting point (i.e., small $\|Z_t - x_0\|$), and more time on the further points. It allows us to adapt the time based on the "hardness" of the target distribution.

**Accelerating the First Hitting Time**    The inverse Gamma distribution above has a heavy tail and occasional causes large hitting time. One way to ensure a bounded hitting time is to derive the conditioned process of Brownian motion given that the hitting time $\tau$ is no larger than a threshold. Specifically, assume $\mathbb{B} : dY_t = dW_t$ starting from $Y_0 = y_0 < y_{\max}$ and $\tau = \inf\{t \colon Y_t = y_{\max}\}$. Using $h$-transform, we can show that $\mathbb{B}(\cdot \mid \tau \leq T)$ is governed by the following diffusion process:

$$\mathbb{B}^T := \mathbb{B}(\cdot \mid \tau \leq T) : \quad dY_t = \nabla_y \log\left(1 - F\left(\frac{|y_{\max} - Y_t|}{\sigma\sqrt{T-t}}\right)\right) dt + \sigma d\tilde{W}_t,$$

where $F$ is the CDF of standard Gaussian distribution.

Taking $b(Y_t, t) = \nabla_y \log\left(1 - F\left(\frac{|y_{\max} - Y_t|}{\sigma\sqrt{T-t}}\right)\right)$ in Eq. 22, we can obtain the following Poisson kernel for $\bar{\mathbb{Q}}$:

$$\bar{\mathbb{Q}}(Z_\tau = dx' \mid Z_t = x, Y_t = y) = \Gamma\left(\alpha, \frac{\phi(x'; x, y)}{T - t}\right) \frac{|y_{\max} - Y_t|}{\phi(x'; x, y)^\alpha} dx',$$

where $\alpha = \frac{d+1}{2}$ and $\phi(x'; x, y) = \frac{1}{2}((y_{\max} - y)^2 + \|x' - x\|^2)$, and $\Gamma(\alpha, x)$ is the upper incomplete gamma function. Correspondingly, the hitting time of this new process is $\mathrm{InvGamma}\left(\frac{1}{2}, \frac{(y_{\max} - Y_t)^2}{2\sigma^2}\right)$ truncated on $[0, T]$.