# OpenReview forum: "First Hitting Diffusion Models for Generating Manifold, Graph and Categorical Data"
_NeurIPS.cc/2022/Conference — NeurIPS 2022 Accept_

### Official Review · Reviewer_FUwB · 2022-07-08

**Rating:** 5
**Confidence:** 4
**Soundness:** 3 good
**Presentation:** 1 poor
**Contribution:** 3 good

**Summary:**

The authors propose a generative model based on the simulation of SDEs that terminate at random hitting times.

Section 2 introduces the relevant background for first hitting diffusion processes, providing some simple examples like hitting of the sphere or the hyperbox. Then it exposes the conditioning principle and the h-transform, that can be used to derive the law of diffusions conditioned on hitting events/times. Section 2.3 explains how to fit a parametric SDE to learn a first hitting diffusion process with the desider exit distribution (eq 12). The authors also include interesting comments on the generation of categorical data and cheap pre-simulation of certain trajectories.
Section 3 is composed by two paragraphs that position the proposed method w.r.t the literature.
Section 4 presents compelling numerical results on various domains, like sphere, graphs and segmentation generation. To the best of my knowledge the comparisons are fair and the list of competitors is accurate.


**Questions:**

* The variational principle (Prop 2.8) and the loss L(\theta) are KL minimization in different directions (direct, reverse). Can you clarify the implications? This would help!
* The conditioning principle: how is it used exactly? Is it only for categorical data?
* Gains from early stopping of certain coordinates: isn’t the computational complexity still the same? For example, when performing segmentation, even if certain pixel are “frozen” the score network still needs to process the full image, right?
* Fast presampling of trajectories and rotation. Is this really trivially applicable to other hitting domains? The authors should sketch a discussion (also in Appendix) or modify the claim
* Diffusion models for sampling: lines 28-30: It is a bit confusing in my opinion to claim that finite time diffusion models outperform infinite time (or mixing), as the former are used typically for data generation, while the latter for Bayesian Sampling and/or optimization. In line 35 it is claimed that the method can be used for inference as well. The authors should either clarify or remove the claim.

Typos:
* Line 153: P\theta_0
* Line 309: “Why can stop at hitting time”
* Algorithm 2.3: where is it?


== Post rebuttal comments==
I have read the rebuttal and decided to increase my score slightly. I still think the notation, and the background knowledge required to digest the methodological part of this work to be worth improving. But this is a valid submission.


**Limitations:**

I think the limitations are exposed clearly and fairly.

**Strengths And Weaknesses:**

The strength of the paper is the introduction of a new (to the best of my knowledge) data generation methodologies, suitable for arbitrary domains. Interestingly, classical diffusion models can be seen as a particular case of first hitting diffusion processes. To the best of my understanding the technical derivations are correct and the claims are mathematically valid. The experiments show the practical usefulness of the method.

In my opinion, one of the biggest drawback of the work is linked to the clarity of the exposition. The authors should clarify/simplify the notation. I think for a reader it is be quite hard to follow the paper, because it uses different notation/symbols to identify similar quantities. In particular:

* Q is used for the paths/Ito process absorbed to \Omega.
* Q is used for times \tau
* Q^{... }_{\Omega} for the harmonic measure/ exit distribution (why the notation with both pedix and superscript?)
* Q^{\Pi^*} is the Markov process with the desired exit distribution. Again, comparing with previous notation: Q^{ something}, “something” is sometimes a set, sometimes a measure
* Q^{x}: paths taken conditionally on starting input x (is this explicitly defined?)
* Q_0: some initial measure
* P^{\theta}, \theta is the set of NN parameters!

Second, the authors should not take for granted certain results, and try to have a paper that is as self contained as possible. I understand many results could be taken for granted in other communities, but this is not the case for the Neurips audience, at least in my opinion. In particular:

* the exit distribution on the sphere when the Brownian motion starts from a point z_0 that is not the center. Reference/ proof sketch?
* Eq (7): be explicit about the math of disintegration and measure products
* Eq (8): this is the central element of the paper, and is left uncommented/unjustified. Consider adding in the appendix a full expansion using h- transform (or any other tool)

---

> ### Author Response · Authors · 2022-08-02
> **Author response to reviewer FUwB**
>
> We thank reviewer FUwB for the comments!
>
> **On notations**
>
> Due to the involved mathematics, designing a good notation system is indeed a challenging task of this work. We tried to use superscripts $Q^{a}$ for $a$ that modify or characterize measure $Q$ (such as the target point $x$, target distribution $\Pi^*$, parameters $\theta$), and use subscripts $Q_{b}$ for $b$ that correspond to marginalization operators on $Q$ (such as the marginal law on a time $t$ or when hitting an exit domain $\Omega$). We could explicitly clarify why this convention in the paper although it is not a 100% rule. We will see how we can improve the notation in the version. Please let us know if you have any suggestions.
>
> **On deriving the exit distribution on the sphere**
>
> Thanks! This can be obtained by directly applying the h-transform formula in Eq (3). We will add corresponding details on that.
>
> **On Equ (7) and Equ (8)**
>
> Thanks for the suggestion! We will clarify it and add related references as well as details in the next version.
>
> **On the variational principle**
>
> There is no specific reason to make the KL divergence in Prop 2.8 and $\mathcal{L}(\theta)$ to be the same, as they serve different purposes. In fact, the KL directions in Prop 2.8 are not important and we can switch the direction to get the same result. For example, it is trivially true that $Q^Pi = \arg \min_P KL(Q^\Pi  || P) = \arg \min_P KL(P || Q^\Pi)$ whenever the optimization domain includes $Q^\Pi$. We will clarify this in the paper.
>
> **On the conditioning principle**
>
> The conditioning principle is a technique that allows us to customize the target domain in a very flexible way. It is only used for categorical data in the paper but can be generally useful. Here is the idea: The boolean hitting would exit at a binary vector $x\in\{0,1\}^d$. On the other hand, the categorical data can be represented as one-hot vectors (a binary vector with exactly one non-zero element), which is a subset of binary vectors. We use the conditioning principle to modify the boolean hitting process to ensure that the process exists at a one-hot vector, rather than an arbitrary binary vector so that we can sample categorical data.
>
> **On computational complexity**
>
> Yes. But our finding is that FHD can generate high-quality data with fewer diffusion steps on average, which saves computation. See section 'Hitting time distribution' in section 4.2 and  'Number of diffusion steps' in Appendix A.6.
>
> **On Fast presampling of trajectories and rotation and diffusion models for sampling**
>
> Thanks for the suggestion! We will show more details for the fast presampling and clarify that the comparison is drawn in the setting when we apply both fixed time and infinite time diffusion models to generative models.

---

> > ### Comment · Reviewer_FUwB · 2022-08-08
> > **Thank you for your comments**
> >
> > Dear authors, thanks for the comments.
> > I think this work has received good scores from all other reviewers, and everyone agrees the idea is novel and with a lot of potential which has been shown empirically.
> > I am still not fully convinced about the notation and exposition, but I will raise my score.
> >
> > Cheers

---

### Official Review · Reviewer_wpmu · 2022-07-11

**Rating:** 3
**Confidence:** 4
**Soundness:** 3 good
**Presentation:** 1 poor
**Contribution:** 2 fair

**Summary:**

This paper proposes FHDM (first hitting diffusion model), which differs from diffusion models based on infinite time (e.g., Langevin dynamics) or fixed time diffusion processes (e.g., DDPM); hence it can be used for data from discrete and structured domain by utilizing the exit distribution for learning and inference. The authors also propose a fast sampling algorithm for the rotational symmetric domain.

**Questions:**

- (L84) Equation 2 seems not general since it has time-independent coefficients in drift and diffusion terms. The solvability of SDEs with time-dependent coefficients is a well-known fact and the original fixed-time diffusion models also assume such conditions. Why do you restrict the condition of the coefficients? See [De Bortoli et al., (2021)](https://arxiv.org/abs/2111.07243) for a reference.
- (L86) Why do we need a unique solvability of Equation 2 in the *weak* sense? For the readers who are not familiar with analytic probability theory, the authors need to specify why we can relax the regularity of the coefficients.
- (L143) Where is Equation 15? Many equations have wrong equation numbers hence it is difficult to understand the statement of the manuscript.
- Typos.
    - (L141-L142): To address this problem, $P_{\Omega}^{\theta}$ of general **general** diffusion process
    - (L141-L142): “**backwar**” way
    - (L660, L662, 665 in Appendix A.9): It is a straightforward application of formula (4) → These are results of Equation 3.

**Limitations:**

The authors mentioned the technical limitations of their work.

**Strengths And Weaknesses:**

(Strength)

- (Significance) The proposed method shows good performance compared to the different genetive models on various domains such as sphere and graph.

(Weakness)

- (Originality) The proposed method shares a similar idea, *diffusion bridge*, with [S. Peluchetti (Non-denoising forward-time diffusion, 2021)](https://openreview.net/pdf?id=oVfIKuhqfC). The authors mention that the FHDM can learn the distribution in various domains by leveraging the Doob $h$-transform; however, the background theory of the two approaches seems equivalent although Peluchetti’s paper focuses on the diffusion mixtures.
- (Quality) The concept of *Diffusion Bridge* is classical in the theory of diffusion processes. However, the authors only cover the diffusion processes with the *time-homogeneous* coefficients (see Equation 2). This setting cannot recover the original setting of fixed-time diffusion models which utilize the SDE classes with time-dependent coefficients.
- (Clarity) The paper seems quite technical for general ML audiences while less informative for the experts in the diffusion theory. Furthermore, the reviewer finds out a few typos and has difficulties reading the proof since the statements of the manuscript are disorganized. The author recommends proofreading before the submission.

***Updated Review***

The reviewer has read the authors' responses. The authors addressed my concerns regarding quality. However, the originality and clarity problems remain; hence I will keep my score.

---

> ### Author Response · Authors · 2022-08-02
> **Response to reviewer wpmu**
>
> We thank reviewer wpmu for raising several concerns and below please find our answer.
>
> **On novelty**
>
> The key contribution of this work is to introduce the first-time hitting mechanism as a new approach to learn distribution on structured domains. The h-transform, which both we and Peluchetti, is only a tool that we use to derive the algorithm.
>
> **On quality, i.e., the time-homogeneous coefficients**
>
> We do use time-dependent coefficients in both the drift and diffusion terms, as indicated by the subscript $t$ in $b_t$ and $\sigma_t$.  We can recover the fixed-time diffusion model as a special case as we show in Example 2.4.
>
> **On weak solution**
>
> Because we only need a weak solution because we only care about the probability law of the stochastic processes.
>
> **On Equ 15 and other typos**
>
> Apologize for the index issue. It should be Equ (20) in the Appendix. We will fix those bugs as well as the typos in the next version.

---

### Official Review · Reviewer_hXZa · 2022-07-11

**Rating:** 7
**Confidence:** 2
**Soundness:** 3 good
**Presentation:** 3 good
**Contribution:** 3 good

**Summary:**

This paper introduces First Hitting Diffusion Models (FHDM), a class of generative models that generalize the fixed time diffusion probabilistic models by enabling instance-dependent adaptive diffusion steps (i.e. terminates at a random first hitting time). Leveraging the framework of first hitting diffusion, FHDM can be applied to a wide range of distributions beyond $\mathbb{R}^d$ by specifying the domain $\Omega$, which could be a sphere or a simplex. Learning and inference are achieved by approximating the drift term in the corresponding Ito diffusion process with a neural network, in analogy to the score models in standard DDPMs. Experiments show that FHDM is able to generate high-quality samples for general data distributions.

**Questions:**

1. Is FHDM applicable for image generation tasks or others whose $\Omega$ are not easily specified?
2. Can you provide the example training time of the experiments? It seems that training FHDM could be time-consuming. For example, How long does it take to train the model for the "Segmentation Map Generation" task?

**Limitations:**

The authors mentioned that training overhead is one of the limitations.

**Strengths And Weaknesses:**

- Originality:
The idea is very interesting and looks novel to me. To produce the data distribution, a different framework is adopted to construct the stochastic process, where only the forward process is considered. This could be an interesting topic for future research.
----
- Quality:
The proposed method looks technically sound. However, I am not familiar with the mathematical background of this paper and can not check the details.
----
- Clarity:
The paper is well-written and the figures are illustrative. Notations are clean and rigorous.
----
- Significance:
The proposed method is verified on several different domains (e.g. point cloud, sphere data, and graphs).

---

> ### Author Response · Authors · 2022-08-02
> **Response to reviewer hXZa**
>
> Thank reviewer hXZa for the positive feedback and comments!
>
> **FHDM for general $\Omega$**
>
> Thanks for this point! Currently, we currently focus on four basic domains considered in this paper as examples to demonstrate our method, since they are the most common ones. We think FHDM can be directly applied to image generation, in which case we can sample discrete pixel values using the hitting mechanism. Generalizing FHDM in more implicitly specified domains (e.g., these implicitly specified by a set of equality or inequality constraints) is a very interesting direction to consider.
>
> **Training time for segmentation map generation**
>
> Thanks for the comments. We have the training time comparison between FHDM and a standard diffusion model in paragraph ‘Acceleration by fast sampling’ in Appendix A.6, where we show that FHDM needs comparable training time to a standard diffusion model.

---

### Official Review · Reviewer_Qdu1 · 2022-07-12

**Rating:** 6
**Confidence:** 4
**Soundness:** 3 good
**Presentation:** 4 excellent
**Contribution:** 3 good

**Summary:**

This paper introduces diffusion processes to generate discrete data by using hitting times as the end of the diffusion process rather than a fixed stopping time. The resulting first hitting diffusion models can be optimized by maximum likelihood, and the learning process is equivalent to matching the drift of the conditioned process. The simulation is done by a forward sampling process using the learned drift.

**Questions:**

1. Can you show the novelty of generated samples? Since the generative process lands on isolated points, the cost of learning to generalize to unseen data points can be bigger than operating in continuous spaces.
2. Can this method works well with structured data, such as molecules?
3. What is the approximate number of diffusion steps when the data dimension grows?

**Limitations:**

The authors properly addressed the limitations in their work.

**Strengths And Weaknesses:**

Strengths:
1. This paper proposes a rigorous approach to generating discrete data by introducing the hitting time on the target range as the stopping time. With the help of the h-transform, one can approximate the desired distribution by modeling the drift term.
2. On symmetric spaces, one can pre-sample paths for faster computation with the rotation trick.
3. Empirical results demonstrate better performance on point cloud generation and graph generation.

Weaknesses:
1. The method seems hard to scale to large and complex images.
2. The sampling mechanism sequentially samples pixels in an image. The sequential selection process places a challenging condition (existing pixels) on the succeeding sampling process. The cost to learn all conditional distributions can be huge.
3. There seems to be no mechanism to restrict the hitting time upper bound.

---

> ### Author Response · Authors · 2022-08-02
> **Response to reviewer Qdu1**
>
> We thank reviewer Qdu1 for the feedback!
>
> **On the scaling FHDM for complicated data**
>
> Thanks! The main focus of FHDM is to provide a novel methodology to learn the distribution of structured data. We believe FHDM can be scaled up for very complicated data. Please let us know if there are specific concerns regarding scalibility and we are happy to address them.
>
> **On the cost of learning all conditional distributions**
>
> Indeed, empirically, we observe that FHDM can generate data faster than standard diffusion suggesting that the cost might actually be small. Our intuition is to “fix easier things first”:  the earlier-exit pixels give strong signals to the remaining evolving pixels and guide them to find their proper value more quickly. You can think of it as an autoregressive model whose sampling order is adaptively chosen. See 'Why can stop at hitting time' in section 4.2 for related discussion.
>
> **On mechanism to restrict the hitting time upper bound**
>
> Our empirical results suggest that the hitting time is well upper bounded. In practice, we can early stop and regenerate a new image if the waiting time has been larger than a given threshold as this only alternates the final generation distribution slightly when the threshold is sufficiently large.
>
> **On novelty of generated samples**
>
> We observe that our model generates novel samples (which is also demonstrated by all kinds of metrics we use in the experiment). For all of the 4 domains, we include the generated samples in Appendix A.7.
>
> **FHDM for molecules data**
>
> Sure, since molecules are a composited distribution of categorical and continuous distribution, it is a highly suitable domain to apply FHDM. We will explore this exciting area in future works.
>
> **On the approximate number of diffusion steps when the data dimension grows**
>
> It grows with a logarithmic rate to the data dimensions. Please also see 'Hitting time distribution' in section 4.2 for some practical value.

---

### Meta-Review · Area_Chair_qP1h · 2022-08-26

**Recommendation:** Accept
**Confidence:** Less certain

**Metareview:**

The paper introduces a new approach for generative modeling: a diffusion process is run until it first hits a target set, and then outputs the first point that is hit.

Three reviewers generally praised the originality, technical quality, and empirical results of the paper. They found the idea very interesting and novel, and technically sound. The numerical results were judged to be compelling and fair. One concern was clarity of exposition. There seemed to be two issues: (1) there were more typos and rough edges than expected, (2) more significantly, there was some difficulty in following all details of the main method given the notational complexity and significant amount of mathematical background on diffusion processes. Reviewer FUwB gave a number of concrete suggestions for improvement.

Reviewer wpmu had a negative overall opinion and critiqued the originality, quality, and clarity. On these issues: (1) the quality concern was based on a misunderstanding that was later resolved, (2) the originality concern does not seem justified to the meta-reviewer (it is based on a shared technical tool with a not-yet-published paper), and (3) the clarity concern is similar to those raised by other reviewers (especially FUwB). Overall, the meta-reviewer does not feel that the low score (3 = “reject”) was fully justified.

In summary, overall reviewers found the paper sound and novel, with the main area for improvement being clarity of exposition about diffusion processes; one reviewer considered originality a weakness, but the meta-reviewer did not find this position well justified.


**Award:**

No

---

### Decision · Program_Chairs · 2022-09-14

Accept